# LWLCM: A novel lightweight stream cipher using logistic chaos function and multiplexer for IoT communications

Shahnwaz Afzal[1], Mohammad Ubaidullah Bokhari[1], Mahfooz Alam [2]*, Mohd Shahid Husain[3], Mohammad Zunnun Khan[4], Zubair Ashraf [5]*

1 Department of Computer Science, Aligarh Muslim University, Aligarh, India, 2 Department of MCA, G.L. Bajaj Institute of Technology and Management, Greater Noida, India, 3 College of Computing and Information Sciences, University of Technology and Applied Sciences, Sur, Oman, 4 Department of Information Systems and Cybersecurity, College of Computing and Information Technology, University of Bisha, Bisha, Saudi Arabia, 5 Department of Computer Science and Artificial Intelligence, College of Computing and Information Technology, University of Bisha, Bisha, Saudi Arabia

* ashrafzubair786@gmail.com (ZA); mahfoozalam.amu@gmail.com (MA)

## Abstract

The Internet of Things (IoT) includes vehicles, homes, and integrated sensors and many interconnected physical devices that gather and share data to interact with their environment. Data moving across multiple levels is vulnerable to various security threats, including leaks and unauthorized access. IoT faces significant challenges in balancing strict security with optimal performance metrics such as energy efficiency, throughput, and memory. We present a novel lightweight stream cipher designed to secure IoT communication and address these challenges. The proposed architecture features four main components: a logistic round module that produces 32-bit chaotic outputs; two 80-bit shift registers, LFSR and NLFSR, for key expansion; and multiplexer units to enhance confusion and diffusion. This model improves the randomness and robustness of the keystream, strengthening the cipher against cryptanalytic attacks. An ablation research is performed by methodically eliminating the chaotic map, NLFSR, and multiplexer components to assess their individual effects on encryption/decryption duration, throughput, entropy, and avalanche analysis. Experimental results demonstrate that each component significantly improves the cipher's overall performance and security, hence confirming the architecture's design and also demonstrate that the proposed cipher exceeds the performance of current algorithms, including Grain-128 and RSA-1024, in terms of encryption/decryption time, throughput, and energy efficiency, while maintaining comparable statistical randomness to AES and Trivium. This method achieves an average Shannon entropy of 7.9996, and successfully passing all 15 NIST statistical randomness tests. A subsequent study analyzing the avalanche effect and correlation coefficients reinforces the strength of the encryption. The proposed encryption method, designed for resource-constrained environments, provides efficient and robust cryptographic security to protect IoT data effectively.

**Data availability statement:** All relevant data are within the manuscript. Powered.

**Funding:** The author(s) received no specific funding for this work.

**Competing interests:** The authors have declared that no competing interests exist.

# 1. Introduction

The Internet of Things (IoT) refers to many types of connected or remotely linked objects and devices [1]. It is becoming more popular due to its applications in transportation, communication, business, education, and more. The rise of hyper-connection through IoT allows businesses and individuals to interact from a distance. To promote the Radio-Frequency Identification (RFID) concept, which involves integrated actuators and sensors, Kevin Ashton coined the term "IoT" in 1999. However, the initial concept was introduced in the 1960s, when it was called integrated Internet or universal computing. Ashton applied IoT to improve supply chain functions. The advantages of IoT led to its significant rise in popularity during the second half of 2010. A remarkable surge began with the introduction of wearable technology, smart energy meters, and home automation in 2011. The rapid growth of IoT has enabled organizations to enhance their commercial strategies and market research in many ways, just as automated services introduced by IoT have improved individuals' lives [2].

Unintentional usage, reusing passwords, and the absence of device patches all exacerbated security holes and gave hostile apps unfettered ability to access private information in IoT devices. These inadequate safety measures raise the probability of data being compromised, along with additional risks. Due to low security norms and procedures, most safety experts view IoT as an attractive attack target. Since the eve of 2008, hackers have created a variety of viruses to infect IoT devices [2]. They produced several phishing techniques to cause employees or others to reveal vital information. Due to high-profile hacks, personal devices and corporate workstations frequently experience privacy invasions. IoT devices have limited resources, including power consumption and memory storage, and security in these devices faces several difficulties. Researchers face challenges in today's reality to improve security in constrained situations. Fig 1 depicts the challenges related to IoT security.

Given these circumstances, encryption may be among the most effective ways to guarantee the confidentiality, integrity, and authorization of data traveling via IoT devices [3]. It may also refer to a method of safeguarding information transmitted or stored via the Internet.

## 1.1. Problem formulation

The traditional PC-based encryption solutions need many resources, and they are unsuitable for IoT devices with limited resources. A simplified version of conventional methods might be used to overcome these obstacles and provide secure communication in IoT devices with constrained resources and lightweight cryptography. The following are the main problems in integrating traditional cryptography in IoT devices, as shown in Fig 2 [4]. Classic cryptographic algorithms such as AES, DES, and RSA are not optimal for constrained devices due to their computational expense, larger memory requirements, and energy inefficiency. Lightweight cryptographic primitives consume reduced power, such as PRESENT and LED block ciphers, along with SPONGENT and PHOTON lightweight hash functions, serve as two exemplars. They have incorporated permutation layers, modify S-boxes, and implement padding techniques, all of which require supplementary memory and computing capacity.

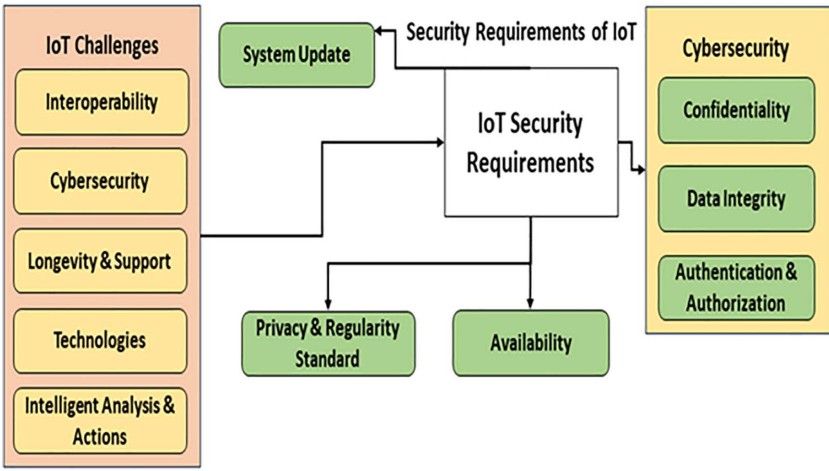

**Fig 1. IoT security challenges.**

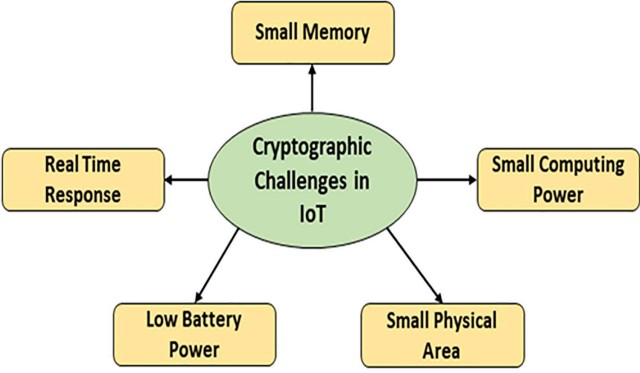

**Fig 2. Key issues with traditional cryptography.**

For instance, even lightweight block ciphers must undergo multiple rounds of substitution and permutation. This indicates that they are ineffective for real-time data encryption on resource-constrained sensor nodes. Lightweight hash-based techniques for authentication and key derivation are effective for data protection and integrity verification; however, they are suboptimal for rapid symmetric encryption and efficient keystream generation during streaming. Conversely, stream ciphers operate exclusively on a single bit or byte at any given moment. This indicates that they can encrypt data rapidly and with minimal memory usage. This indicates their suitability for IoT applications that require minimal power, memory, or CPU resources and have low latency requirements.

Furthermore, existing lightweight cryptographic systems such as Trivium [5], MICKEY [6], and Grain series ciphers [7,8] are a few of the primary stream ciphers that may exhibit insufficient resistance to modern attack vectors or fail to achieve an acceptable balance between security and performance. Stream ciphers designed for resource-limited environments must provide low latency and minimal resource usage while preserving high unpredictability, nonlinearity, and cryptographic strength. Achieving this equilibrium remains difficult. A significant demand exists for lightweight encryption models that minimize computing complexity while incorporating robust diffusion, dynamic key evolution, and improved entropy. Over time, the decoding science has made significant progress, and it has now been shown that certain lightweight

stream ciphers are dangerous [9]. Extensive key search attacks could be more effective against Grain v1. To address these challenges, there is a pressing need for a novel lightweight stream cipher architecture that:

- Ensures robust security through high nonlinearity, resistance to statistical and algebraic attacks, and strong keystream randomness.

- Operates efficiently on constrained IoT devices by minimizing computational overhead, memory usage, and energy consumption.

- Improves upon existing models such as Grain, Trivium, and other chaos-based designs by offering better entropy, faster encryption, and full NIST randomness compliance.

- Incorporates dynamic and adaptive cryptographic structures, such as chaotic maps and multiplexer-based nonlinear mixing, to enhance confusion and diffusion without compromising performance.

## 1.2. Contribution

This work integrates the nonlinear Feedback Shift Register (NLFSR), Linear Feedback Shift Register (LFSR), and the logistic chaos function to provide resistance against different kinds of attacks. Cryptographers are beginning to embrace chaotic cryptography as a revolutionary theory of cryptography. Chaotic cryptography is widely used in neural networks, economics, and secure communication [10,11]. However, only some studies have attempted to apply chaotic structures to lightweight cryptography. Researchers have been examining structurally modified chaotic maps to address the issues associated with the standard logistic map, including short periodicity in digital implementations and diminished Lyapunov exponents. The study referenced in [12] employs a multi-parameter hybrid chaotic map to enhance the sensitivity and reduce the unpredictability of image encryption. The research in [13] employed a Logistic-Sine connected system to enhance resistance against statistical attacks. This study deliberately selects the standard logistic map due to its minimal computational requirements, consistent parameter responses, and compatibility with LFSR and MUX logic. This is ideal for encrypting concise IoT SMS communications. The subsequent phase involves examining adaptive or hybrid chaotic systems to determine their potential to further enhance cryptography. In the past, many studies have been conducted on the integration of chaotic systems with stream ciphers. For instance, NLFSR Kumari & Mondal in [14], and Wang et al. [15]. The majority of existing research has focused on enhancing key sensitivity and fortifying PRNGs against attacks. Conversely, the LWLCM model is an adaptive amalgamation that optimizes resource utilization in response to its environment. It was specifically designed for IoT edge devices.

Using a thorough analysis of other lightweight stream ciphers, this work combines a chaotic system with NLFSR and LFSR to create a novel chaos-based lightweight stream cipher system supported by multiplexer units to introduce enhanced confusion and diffusion properties. Avalanche analysis and NIST procedures are applied to assess the lightweight stream cipher performance, being able to validate the encryption's features. The results demonstrate the superior performance of the more straightforward encryption used here. LWLCM's design is the same as that of Grain v1, including a multiplexer and logistic chaos as a logistic round module to increase security. The following have been highlighted as this paper's main contributions:

- LWLCM is an innovative stream cipher that employs a chaotic pseudo-random number generator based on the Logistic Map to amalgamate NLFSR and LFSR for keystream generation. The chaotic component enhances nonlinearity and improves statistical unpredictability, distinguishing it from conventional PRNGs.

- Due to our modular architecture, we can conduct ablation experiments to evaluate the impact of each component on the quality of the keystream and the execution speed. This illustrates the significance of each component.

- This study highlights the enhancement of statistical security through Shannon entropy and the reduction of correlation coefficients, introducing a novel lightweight stream cipher, LWLCM, tailored for IoT environments.

- The main aim of this work is to produce extremely random ciphertexts with optimal unpredictability, thus enhancing resistance to statistical and linear attacks.

- Encryption is more resistant to attacks, such as linear and differential cryptanalysis, when complex non-linear Boolean functions that involve XOR, AND, and multiplexer operations are used. These functions improve the cipher's security against sophisticated cryptanalytic techniques by adding a level of non-linearity and complexity that is challenging to reverse. With a maximum nonlinearity of 19, the filtering function is a symmetric Boolean computation with eight variables and seven orders.

- An extra degree of complexity and security is added by using multiplexers to regulate the selection of operations based on the state of particular bits. This design decision strengthens the cipher's resistance to algebraic and fault attacks by making any straight linear approximation more difficult.

- Experimental analysis demonstrates that the proposed LWLCM cipher achieves its primary design objective, enhancing randomness and diminishing statistical correlation in ciphertext generation through efficiency. Across various plaintext sizes, LWLCM consistently produces entropy values near the optimal value of 8.0 and correlation coefficients nearing zero. These findings indicate that the cipher can yield significantly random and statistically independent outputs, which are essential indicators of robust encryption. The superior performance in statistical security measures justifies the trade-off, despite its encryption and decryption rates being somewhat higher than those of AES-256 and Trivium.

This paper is divided into several Sections. Section 2 includes the literature review, while Section 3 describes the design specification. The Design consideration is defined in Section 4. At the same time, the proposed work LWLCM model and algorithmic template, is discussed in Section 5. The results and discussion are covered in Section 6 with complete observation, followed by security analysis, which is covered in Section 7. The statistical test is shown in Section 8, and the limitations of the proposed work have been discussed in Section 9. Finally, the conclusion and future directions are covered in Section 10.

## 2. Literature review

Recent years have seen extensive research on stream cipher designs and chaotic systems due to the high demand for encryption that is both lightweight and secure. In this section, we review the existing literature and highlight areas for improvement. Next, we discuss the need for the LWLCM architecture. In [14], it was claimed that many vulnerabilities, such as spoofing, jammer attacks, and other unauthorized access methods, compromised the security of user information. Indeed, there are options available to help individuals protect their IoT devices by employing different security measures. Additionally, the author introduces a new method for fast and highly secure image encryption in [15], combining byte-level scrambling with an improved variant of the Trivium cipher. Permutation sequences based on the Hénon chaotic map increase unpredictability and security. The methodology was tested using over one hundred color photos across multiple performance and security evaluations. In [16], the author developed a technique to enhance the performance of IoT systems operating over 5G networks by reducing interference. It uses pre-coding methods with spatial filtering and beamforming to lessen interference in dense 5G-IoT deployments. This approach effectively mitigates multi-user interference and boosts the reliability and throughput of IoT devices in 5G.

Cryptographic algorithms have improved security and privacy in various IoT applications, such as smart homes and cities, electronic medical devices, health monitoring, object tracking, and monitoring. IoT systems comprise readers, actuators, microprocessors, sensors, RFID tags, and other hardware with various processing capacities. The intricate architecture of the IoT, which entails the integration of numerous, frequently anonymous, and perhaps unreliable devices,

protocols, and resources, presents serious security and privacy issues. For this reason, in [8] et al., the author designed a stream cipher for hardware systems where memory, power, and gate count are strictly limited. Its nonlinear filter function and two shift registers form the foundation of its architecture, which guarantees effective operation even in these limited circumstances. Grain is also scalable, meaning adding more hardware resources can result in faster performance. Due to its effectiveness, scalability, and security, the cipher is a desirable alternative for encryption in hardware environments with constrained resources. The authors of [9] et al. developed Grain-128, a small and effective stream cipher, especially for settings with extremely constrained resources regarding chip space, power consumption, and gate count. Grain-128, optimized for hardware implementation, allows an IV of 96 bits and a key size of 128 bits. One linear and one nonlinear register and an output function are incorporated into the design to ensure simplicity and ease of implementation. Because of its simple technique, Grain-128 is a practical and efficient solution for safe communication if hardware is restricted. The authors of [6] et al. proposed a hardware-oriented synchronous stream cipher with an area-efficient design that balances performance and speed. Three feedback shift registers are used in this encryption; each has an 80-bit IV and 80-bit key size. The cipher provides robustness and encryption strength with an internal state of 288 bits. The design is versatile in performance and optimizes hardware resources, making it appropriate for various hardware configurations with different needs. The excessive number of rounds and the requirement for a diffusion operation inside the round function in [17] make classical block ciphers computationally expensive. Block ciphers currently in use have a minimum of four rounds, such as the Hummingbird2 cipher. That kind of expense should not be necessary for specific emerging systems. Lately, there has been a surge in the creation of new, efficient encryption algorithms that need fewer resources and have reduced latency. The author of [18] suggests a novel method for image encryption using chaotic logistic maps to satisfy the criteria of safe picture transit. The proposed picture encryption method employs an external 80-bit secret number and two chaotic logistic models. To fortify the cipher against intrusions, the secret key is modified after encryption every block of sixteen pixels in the picture. An LFSR and a specific nonlinear filter function were used by [19] to build the PRNS. The system selects a square picture first, then applies column-to-row and circular shift operations. Arnold's transformation was then used to warp the image further. Finally, XOR the keystream with the jumbled picture to get the cipher image. The authors of [20] presented the use of the new chaotic map known as the composite logistic sine map (CLSM) and the Secure Hash Algorithm-256 (SHA-256). With the CLSM, the necessary PRNSs are generated. With this approach, the SHA-256 value distorts the pixel value, and the PI's pixels are permuted depending on the PRNS. By executing extended Arnold change, arbitrary circular bit-shift, and bit XOR using random values derived from the aperiodic logistic chaotic map, the author's technique in [21] generated the effects of bit scrambling and bit replacement. Strong encryption was only one of their objectives; another was restoring a picture damaged by noise or enemy attacks. Using the sqrt function and division operation, the random sequence's chaotic function produces each bit of the random sequence. In [22] et al., the authors presented a novel technique for rearranging the pixels in a color image. Subsequently, the study presents an image encryption algorithm that utilizes Arnold's cat map and a logistic map to protect color images. The method is suited for real-time application implementation because of its high level of security and resilience against many forms of noise, data loss, and cryptographic attacks, as demonstrated by the results and several security tests. The authors of [23] et al. described an image encryption method that uses hybrid chaotic maps, namely the 2D-Tent Cascade Logistic Map (2D-TCLM) and the Logistic Map. According to the author, the process performs better than current methods in most essential aspects, making it ideal for real-time image transmission across unreliable public channels. To provide highly randomized and unexpected cryptographic keys for cutting-edge algorithms like AES, SIMON, DES, SPECK, 3DES, and PRESENT Ciphers, Jawed et al. introduced a unique technique [24]. This algorithm makes use of sample entropy and the whale optimization algorithm. Based on comparative evaluations, the author asserts that SampEn-WOA-generated cryptographic keys show potential strength for various cryptographic algorithms, making them a worthwhile option for protecting IoT data. In [25], variable modification in ciphertext bit-length throughout intermediate encryption rounds defines a novel block cipher-based cryptosystem, bFLEX-γ. The key scheduling system improves cryptographic strength via a

randomized diffusion process that employs an auxiliary vector produced by a key-crossing technique in conjunction with an LFSR.In [26], the author introduces chaos theory-based image encryption using an upgraded Grain-128 cipher. The Hilbert curve and Hénon map permute the unaltered image's pixels and bits to reduce pixel correlation. Modifying the Boolean function and lowering the internal state size improve Grain-128. The encrypted image is created by XORing the changed cipher with the randomized picture data. These changes aim to improve encryption and decryption. As we have seen in Table 1, the summarized comparative analysis has been presented.

**Table 1. A comparative analysis of existing work.**

| Author | Purpose | Technique Used | Strength | Limitation |
|---|---|---|---|---|
| Meng et al., [16] | Address security challenges in consumer IoT smart homes | Voice liveness detection, smart home platform testing | Real testbed validation; novel voice interface security | Focused mainly on voice security, not general encryption |
| Ali et al., [17] | Develop a fast and secure image encryption using Trivium with the Henon map. | Modified Trivium + Henon Map + byte scrambling | Fast, energy-efficient, resistant to statistical attacks | Only image encryption, specific to the Henon map |
| Noura et al., [18] | Propose a one-round cipher for lightweight multimedia IoT | Single-round cipher with dynamic key, permutation, PRMs | High speed, low error propagation, good for multimedia | Limited to multimedia and one-round operation |
| Mfungo et al., [15] | Propose image encryption using chaotic maps and fuzzy numbers | Logistic/Sine maps + fuzzy logic + Henon map | High key sensitivity, statistical resistance | A complex system with multiple chaotic components |
| Deb & Bhuyan, [19] | Design a chaos-based encryption using a nonlinear filter LFSR | LFSR + nonlinear filter + logistic-tent map | Efficient PRNG, high throughput, cryptanalysis resistance | Bit-level LFSR may face latency for large images |
| Suman et al., [20] | Propose secure image encryption using CLSM and SHA-256 | Composite Logistic Sine Map + SHA-256 hashing | High keyspace, suitable for low-power devices | Needs large nonce/key setup; Entropy alone is not enough |
| Li et al., [21] | Design real-time image encryption with a chaotic map and bit shift | Aperiodic chaotic map + random cycling bit shift | Real-time capable, robust to noise and attacks | May not handle highly degraded inputs well |
| Jawed et al., [22] | Enhance cryptographic key generation using Sample Entropy and WOA | Sample entropy + Whale Optimization Algorithm | Highly random keys, applicable to standard ciphers | Focused on key generation, not full encryption pipeline |
| Das et al., [23] | Propose a flexible, lightweight block cipher with a dynamic block length for enhanced security. | SPN network, dynamic block resizing, key-cross scheduling using PDF, LFSR-based key updates | Supports confusion and diffusion via S-box and P-box, dynamic key generation, resistant to multiple attacks (linear, differential, related-key, interpolation) | Designed for 32-bit blocks; scalability and performance on extremely constrained devices are not fully explored. |
| Deb et al., [24] | Propose a chaos-based image cryptosystem by modifying Grain-128 to improve encryption efficiency and randomness. | Hilbert curve + Hanon map for permutation; modified Grain-128 with reduced state size and custom Boolean function; XOR with randomized image data | High nonlinearity in Boolean function; improved encryption-decryption efficiency; strong statistical randomness; effective against correlation | Focused only on image encryption; relies heavily on specific chaotic maps; impact on general-purpose data not analyzed. |
| Jaleel et al., [25] | To enhance the performance of IoT systems operating over 5G networks by reducing interference. | Pre-coding techniques based on spatial filtering and beamforming to mitigate interference in dense 5 G-IoT deployments | Effective in reducing multi-user interference Enhances the reliability and throughput of IoT devices in 5G | No hardware or energy analysis for constrained IoT nodes. Focused mainly on physical layer optimization; lacks end-to-end security considerations |
| Kumar et al., [26] | To develop an efficient and secure image encryption scheme suitable for real-time applications and low-resource environments | Novel pixel shuffling technique | Strong resistance against noise, data loss, and cryptanalytic attacks | Focused only on color images, not generalized for other data types |
| Kumar et al., [27] | To enhance the confidentiality of multimedia images during transmission over insecure public networks | Hybrid chaotic system using the Logistic Map and 2D-TCLM | Strong performance against standard cryptographic attacks | Limited to image encryption (no discussion of scalability to video or text) |

Despite extensive research on chaos-based encryption concerning picture encryption and neural networks, its implementation in lightweight stream cipher systems has not garnered sufficient attention. Most prior chaos-integrated models either depend on high-dimensional chaotic maps unsuitable for constrained IoT environments or concentrate on multimedia data. The proposed encryption employs a one-dimensional logistic map alongside LFSR and NLFSR components, resulting in a lightweight and computationally efficient architecture. This combination ensures real-time performance and low memory usage while still enabling us to utilize the erratic nature of chaotic dynamics. Furthermore, the integration of MUX-based nonlinear mixing enhances diffusion and bolsters resilience against conventional stream cipher attacks, hence distinguishing our work from typical chaos-based systems.

## 3. Design specification

The LWLCM model was derived from the Grain family of stream ciphers. The LWLCM's inner state is split between two connected feedback registers for shifting (FSRs) and a stochastic round module, as seen in Fig 3. It comprises an 80-bit NLFSR and an equal-sized 32-bit logistic round module. The logistic round module is also part of the state initialization process, and shift registers provide input to a nonlinear output function, similar to Grain [7]. Section IV will cover the decision to choose this particular design.

### 3.1. Components

The algorithm's primary components are a Logistic round module, two 80-level FSRs, and four multiplexers. LWLCM is composed of 192 bits that are distributed over LFSR, NLFSR, and the Logistic round module. Both the registers LFSR and LFSR are specified over the Galois field $(2^{80})$, and the contents of both registers are designated by $(p_0, p_1, \ldots \ldots, p_{80})$ and $(\beta_0, \beta_1, \ldots \ldots, \beta_{80})$ respectively, as illustrated in Fig 3 and the Logistic round module's contents are designed by $(f_0, f_1, \ldots \ldots, f_{32})$ and updated by the logistic chaos function.

 **3.1.1. Logistic chaos function.** Cryptography has made extensive use of the chaos theory of nonlinear research. Chaotic systems offer beneficial cryptographic properties, including long-term unpredictability, pseudo-random behavior, and extreme sensitivity to initial conditions. These properties are connected to modern cryptography's diffusion and confusing ideas [28]. An approach to chaotic digitization is proposed to address the problem of limited accuracy. Here is how to express the logistic chaotic map given in Equation 1.

$$x_t = \alpha\, x_{t-1}\left[1 - x_{t-1}\right]; \ \alpha \in (3.6,\ 4) \ \& \ x_{t-1} \in (0,1] \quad t = 1, 2, \ldots \tag{1}$$

The initial parameters, $\alpha$ and $x_0$, are variables to maintain uncertainty and security. Rather, they are generated dynamically during the initialization process as follows:

- **Initial Seed $x_0$.** Generated using a portion of the secret key. Specifically, the ASCII values of the first few characters of the key are normalized to the range (0,1) as:

$$x_0 = \frac{1}{K}\sum_{i=1}^{K}\frac{\mathrm{ord}\,(k_i)}{256} \tag{2}$$

where $k_i$ is the $i^{th}$ character of the key and $K$ is the key length.

- **Logistic Parameter $\alpha$:** Derived using a combination of IV and key entropy using:

$$\alpha = 3.6 + \left(\sum_{j=1}^{L}\frac{\mathrm{ord}\,(iv_j)}{256}\,mod\ 0.4\right) \tag{3}$$

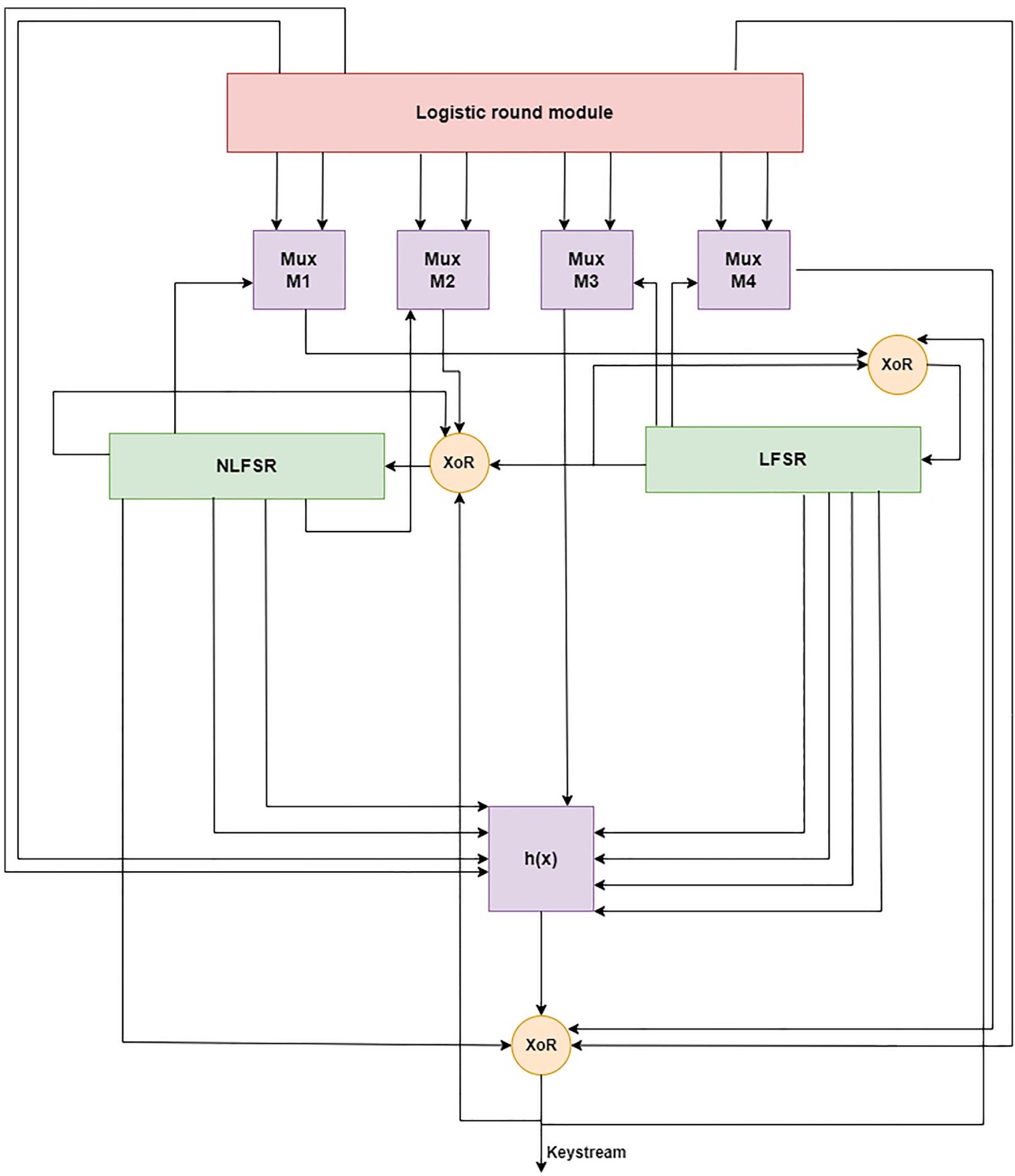

**Fig 3. Overview of LWLCM.**

This ensures that $\alpha \in (3.6, 4.0)$, maintaining chaotic behavior while introducing sensitivity to IV changes. Fig 4 shows that the chaotic map has more variability across the interval $3.6 \leq \alpha \leq 4$. However, we have to consider that digital degradation can significantly affect the performance of the logistic map, especially in digital systems where absolute numbers are represented with finite precision. Quantization and rounding errors are minor errors that can compound over iterations and cause deviations from the expected chaotic behavior; in specific scenarios, this degradation can result in a loss of chaos, where the system turns into less practical applications that require strong randomness, such as cryptography. Limited accuracy in mathematics can also result in overflow or underflow mistakes, which further ruins the chaotic structure of the logistic map [29]. Due to its high sensitivity to initial conditions, the map is especially susceptible to digital degradation, where even small adjustments can result in considerable differences in output. This decrease in randomness may jeopardize the logistic map's security and efficacy in encryption applications, as small mistakes can propagate and amplify across iterations, decreasing the map's practical dependability.

In the proposed LWLCM system, the chaotic parameters $\alpha$ and $x_0$ are neither stored nor transmitted across the communication channel. Both parameters are generated in real time on the receiver's side, utilizing a shared secret key and an IV. The recipient can independently ascertain $\alpha$ and $x_0$, as the key and IV are provided in advance via a secure method for identification verification. This indicates they are not required to store or transmit these numbers in multiple locations. Attackers cannot recover $\alpha$ or $x_0$ without the secure key and IV, rendering it difficult for them to retrieve the parameters. This also reduces the volume of communication required, which is crucial in IoT environments where resources are limited.

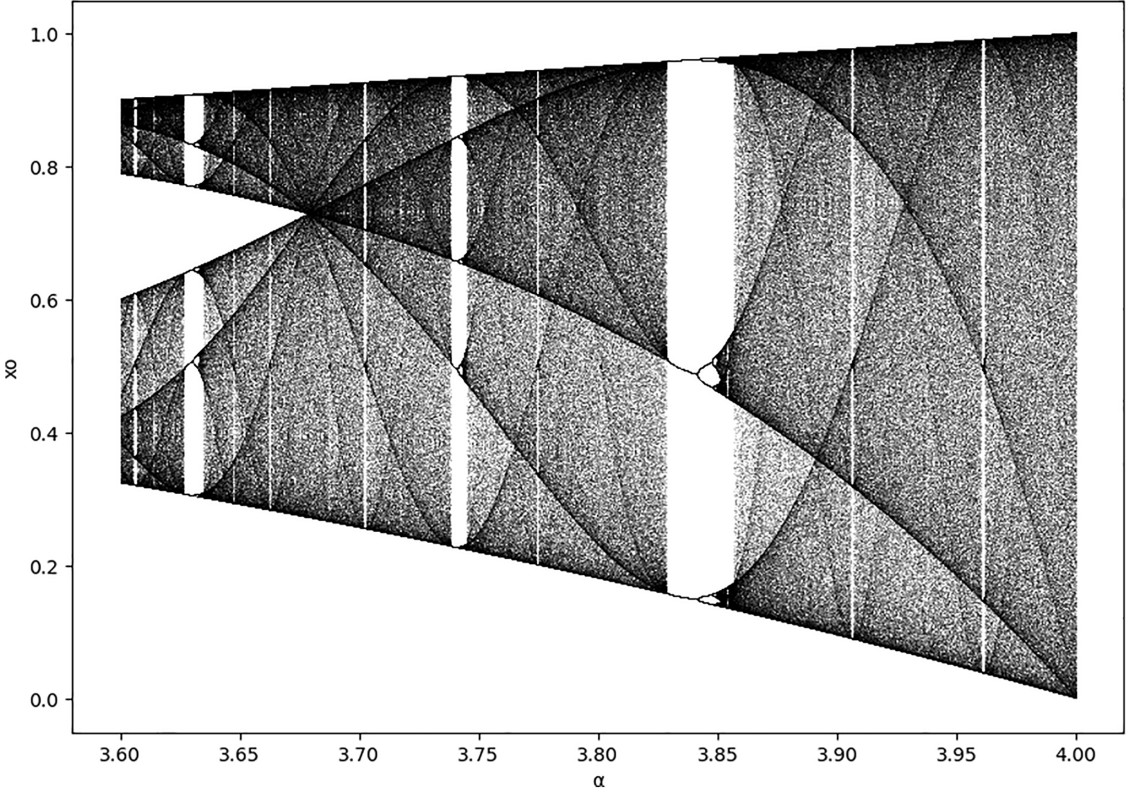

**Fig 4. Bifurcation diagram of logistic map.**

### 3.1.2. LFSR.

An LFSR is a sequential shift register in which its previous state dictates a linear function of its feedback. Except for the first flip-flop, which receives input from a linear function based on the prior state, the configuration comprises a sequence of flip-flops, each receiving input determined by the output of its antecedent. XoR operations on designated bits of the register often fulfill the role. LFSRs have numerous benefits in cryptography and computing contexts because of their simplicity and efficiency, which are greatly valued. Possessing fewer logic gates than alternative pseudo-random number generators (PRNGs), they exhibit considerable efficiency for hardware implementations, characterized by diminished circuit complexity and lower power consumption. Fundamental XOR and shift operations enable LFSRs to perform swiftly, rendering them suitable for high-speed applications such as cryptographic stream ciphers like Grain-128a and Trivium. They effectively generate pseudo-random sequences, which are valuable for Monte Carlo simulations, data obfuscation, and cryptographic key creation. LFSRs, constructed from a fundamental polynomial, yield an extended periodic sequence before repetition, which is crucial for security applications because it generates a maximum-length sequence of $2^{n-1}$. Their reduced processing demands render them more power-efficient than linear congruential generators among conventional pseudorandom number generators. Key components of stream ciphers, particularly A5/1 and A5/2, utilized in GSM encryption, comprise LFSRs. Frequently used alongside NLFSRs, they enhance security in lightweight cryptographic systems.

The 80-bit LFSR utilized in the proposed encryption is constructed with a primitive polynomial defined over the Galois field GF($2^{80}$), which is used by Grain v1 given by Equation 2, ensuring a maximum period of $2^{80}-1$, hence providing optimal periodicity and cryptographic strength. This guarantees that, to prevent predictability in the keystream, the LFSR state will not repeat until the entire cycle is finished. The taps employed are chosen based on recognized elementary polynomials documented in literature and cryptographic standards [30]. The primitive polynomial and state update function of the LFSR are illustrated by Equations 4 and 5.

$$f(m) = 1 + m^{18} + m^{29} + m^{42} + m^{57} + m^{67} + m^{80} \tag{4}$$

$$p_{i+80} = p_{i+62} + p_{i+51} + p_{i+38} + p_{i+23} + p_{i+13} + p_i + M_1 \tag{5}$$

### 3.1.3. NLFSR.

NLFSR, a version of LFSR, utilizes non-linear feedback mechanisms rather than basic XOR-based linear processes, making it more robust against cryptanalysis. NLFSRs utilize AND, OR, and majority logic to incorporate non-linearity, unlike LFSRs that produce predictable sequences via linear recurrence relations, thus markedly improving security. NLFSRs are ideal for cryptographic applications, such as lightweight stream ciphers like Grain-128a and Trivium, because of their non-linearity, which enhances resistance to algebraic and correlation attacks. Improved statistical randomization and increased unpredictability ensure enhanced key generation and encryption security. The 80-bit NLFSR used in the proposed work enhances its resilience against correlation attacks through the use of several nonlinear tap functions and cross-product terms. While NLFSRs do not conventionally employ primitive polynomials, their nonlinear feedback functions are meticulously designed to augment nonlinearity and complexity, hence boosting the unpredictability and diffusion characteristics of the output keystream. The feedback polynomial used by NFSR, which has an 80-bit width, is a modified version of Grain [7]. The following is the update function of the NLFSR is given by Equation 6:

$$\begin{aligned}
\beta_{i+80} = {} & p_i + \beta_i + \beta_{i+63} + \beta_{i+60} + \beta_{i+52} + \beta_{i+45} + \beta_{i+37} + \beta_{i+33}\beta_{i+28} + \beta_{i+21} + \beta_{i+15} + \beta_{i+19} + \beta_{i+63}\beta_{i+60} + \beta_{i+37}\beta_{i+33} \\
& + \beta_{i+15}\beta_{i+19} + \beta_{i+60}\beta_{i+52}\beta_{i+45} + \beta_{i+33}\beta_{i+28}\beta_{i+21} + {} + \beta_{i+63}\beta_{i+45}\beta_{i+28}\beta_{i+9} + \beta_{i+60}\beta_{i+52}\beta_{i+37}\beta_{i+33} \\
& + \beta_{i+63}\beta_{i+60}\beta_{i+2}\beta_{i+15} + \beta_{i+63}\beta_{i+45}\beta_{i+60}\beta_{i+52}\beta_{i+37} + \beta_{i+33}\beta_{i+28}\beta_{i+21}\beta_{i+15}\beta_{i+9} \\
& + \beta_{i+52}\beta_{i+45}\beta_{i+33}\beta_{i+37}\beta_{i+28}\beta_{i+21} + M_2
\end{aligned} \tag{6}$$

**3.1.4. Multiplexer unit.** Cryptography utilizes a 2×1 multiplexer (MUX) to select one of two input signals according to a control bit, enhancing security, efficiency, and diversity. Stream ciphers, substitution-permutation networks (SPNs), PRNGs, and fault-tolerant cryptographic circuits are extensively employed. By altering feedback routes, a 2×1 multiplexer enhances non-linearity and unpredictability in Grain and Trivium stream ciphers, increasing their resistance to cryptanalysis. It facilitates dynamic modifications of block ciphers, enhancing dispersion and confusion. It also safeguards against side-channel attacks by concealing power and timing analysis vulnerabilities. It's hardware efficiency renders it appropriate for lightweight IoT cryptography since it utilizes minimal logic resources and energy while accelerating encryption. MUX-based redundancy enhances the fault tolerance of cryptographic hardware, guaranteeing dependable and secure encryption. Contemporary cryptographic systems depend on the 2x1 multiplexer due to its ability to lower computing expenses, improve resilience to attacks, and facilitate regulated randomization, hence augmenting security. The $M_1$, $M_2$, $M_3$, and $M_4$ multiplexers make up the 2×1 multiplexer unit [31]. The input bits for the $M_1$ originate from $f_{i+4}$ and $f_{i+8}$ bits of the Logistic Round Module; the selection bits for the $M_1$ come from $\beta_{i+19}$ bit of the NFSR, as shown by Equation 7. While the input bits for $M_2$ come from $f_{i+10}$ and $f_{i+16}$ bits of the Logistic Round module, the selection bits for $M_2$ come from $\beta_{i+27}$ bits of the NFSR, as shown by Equation 8. In the same way, the input bits for $M_3$ come from $f_{i+20}$ and $f_{i+24}$ bits of the Logistic round module, and the selection bits for $M_3$ come from $p_{i+11}$ bit of the LFSR, as shown by Equation 9. The selection bits for $M_4$ come from $p_{i+79}$ bits of the LFSR, whereas the input bits come from $f_{i+30}$ and $f_{i+31}$ bits of the Logistic round module, as shown by Equation 10. A two-choice multiplexer that accepts $f_1$ and $f_2$ as input signals and stands for the selection signal is defined as MUX $(f_1, f_2, s)$.

$$M_1 = MUX(f_{i+4}, f_{i+8}, \beta_{i+19}) \tag{7}$$

$$M_2 = MUX(f_{i+10}, f_{i+16}, \beta_{i+27}) \tag{8}$$

$$M_3 = MUX(f_{i+20}, f_{i+24}, p_{i+11}) \tag{9}$$

$$M_4 = MUX(f_{i+30}, f_{i+31}, p_{i+79}) \tag{10}$$

**3.1.5. Logistic round module.** The Logistic Circular Module is easy to use and requires fewer hardware resources. The obfuscation and disturbance component for the entire network in the LWLCM is, therefore, chosen to be the chaotic sequence. Upon system initialization and digitization, the multiplexer unit selects and extracts each of the following bits: $f_{i+4}$, $f_{i+8}$, $f_{i+10}$, $f_{i+16}$, $f_{i+20}$, $f_{i+24}$, and $f_{i+30}$, $f_{i+31}$. A 32-bit Logical Chaotic module is thus created. As a result, the data from the output function, filter function, LFSR, and NLFSR need to be more apparent.

**3.1.6. Filter function.** The data in the two shift registers and the Logistic round module represent the cipher's state. This state gives nine variables to the Boolean function $h(\gamma)$. This filter function was selected due to its balance, first-order correlation immunity, and algebraic degree of eight. The nonlinearity for these functions is precisely 19, which is the maximum possible value. Both the register and the logistic chaotic module provide input. We define the function by Equation 11.

$$\begin{aligned} h(\gamma) = \ & \gamma_8 + \gamma_1 + \gamma_4 + \gamma_0\gamma_3 + \gamma_2\gamma_3 + \gamma_3\gamma_5 + \gamma_0\gamma_1\gamma_2 + \gamma_0\gamma_2\gamma_3 + \gamma_0\gamma_2\gamma_4 + \gamma_1\gamma_2\gamma_4 + \gamma_2\gamma_3\gamma_4 \\ & + \gamma_2\gamma_3\gamma_5 + \gamma_0\gamma_1\gamma_2\gamma_3 + \gamma_0\gamma_1\gamma_2\gamma_4 + \gamma_0\gamma_1\gamma_3\gamma_4 + \gamma_1\gamma_2\gamma_3\gamma_4 + \gamma_0\gamma_1\gamma_2\gamma_4\gamma_5 \\ & + \gamma_1\gamma_2\gamma_3\gamma_5\gamma_6\gamma_7 + \gamma_0\gamma_2\gamma_3\gamma_4\gamma_5\gamma_6\gamma_7 \ + \gamma_1\gamma_2\gamma_3\gamma_4\gamma_5\gamma_6\gamma_7 + \gamma_0\gamma_1\gamma_2\gamma_3\gamma_4\gamma_5\gamma_6\gamma_7 \end{aligned} \tag{11}$$

Where $\gamma_0$, $\gamma_1$, $\gamma_2$, $\gamma_3$, $\gamma_4$, $\gamma_5$, $\gamma_6$, $\gamma_7$, $\gamma_8$ correspond to the tap points $p_i$, $p_{i+25}$, $p_{i+46}$, $p_{i+64}$, $\beta_{i+63}$, $\beta_{i+79}$, $f_{i+15}$, $f_{i+20}$, and $M_3$. The filter function is concealed using bits $\beta_i$, $M_4$, and $f_{i+4}$ to generate the output function.

**3.1.7. Output function.** The keystream bit $w_t$ at the time step $t$ is computed by a nonlinear mixing of bits derived from the chaotic sequence $f_t$ derived from the chaotic module by Equation 1, the NLFSR state $\beta_t$, the multiplexer unit $M_t$ derived from Equation 10, $h_t(\gamma)$ is the filter function derived by Equation 11, and $\oplus$ XOR operation. Formally, the LWLCM's output function is described by Equation 12:

$$w_t = \sum_{j \,\in C} h_t(\gamma) \oplus \beta_t[a_i] \oplus] f_t[4] \; \oplus M_t[4]$$

(12)

where C = {2,15,36,45,64,73,79}. This framework ensures that the keystream comprises state bits and dynamic random input that evolves. This renders it exceedingly difficult to decipher and resistant to linear cryptanalysis. $w_t$ is a robust output function due to the incorporation of a multiplexer, NLFSR, and chaotic map, which enhance its unpredictability and resistance to decryption attempts. Before any key creation occurs, the registers and the logistic round module must be initialized with the key, IV, and logistic chaos functions. Let's say that ki stands for the key bits, where $0 \le i < 79$, and $IV_i$ stands for the IV bits, where $0 \le i \le 63$. The following describes how the key is initialized. To fill up the remaining bits of the LFSR, $p_i = 1$, $64 \le i \le 79$, load IV, $p_i = IV_i$, $0 \le i \le 63$, into the first 64 bits of the LFSR first. Then, insert the key bits, $\beta i = ki$, $0 \le i \le 79$, into the NFSR. Using the logistic chaos function, the logistic round module was initialized with $\alpha \in [3.6,4]$ and $x_0 \in (0,1)$. Next, 160 clock cycles of the cipher are performed without producing a running key that looks like grain. Instead, the filter function's output $w(\gamma)$ is transmitted again and combined with the LFSR and NFSR inputs, as illustrated in Fig 5.

## 4. Design consideration

Given that the maximum period of $2^{80-1}$ is guaranteed, the Grain LFSR, based on a primitive polynomial, is regarded as an LFSR of size 80 bits. Hardware efficiency and low complexity characterize LFSR. The bit generated by the multiplexer $M_1$ masks the LFSR to improve the key's unpredictability and boost its security against various attacks. The 80-bit wide NFSR uses a modified version of Grain's g feedback polynomial, which is still considered unbreakable after years of rigorous cryptanalysis. NFSR is essentially a filter rather than a true NFSR matching remark for the Grain family because of the masking bit from LFSR and Multiplexer $M_2$, leaving the possibility that NFSR happened during keystream formation. This state gives sixteen variables to an NFSR Boolean function. This function was chosen because of its algebraic degree of five, immunity to first-order correlation, and balanced character. The maximum nonlinearity that can be achieved for these functions is precisely 10. The logistic round module is the LWLCM's third 32-bit building component. The logistic chaos function, which is quick, simple to employ in small areas, and nonlinear, initiates it. The starting value of the logistic chaos function and the values of $\alpha$ are generated using Equation 3, which makes predicting various assault types very difficult. One of the primary areas of concern in the layout of a stream cipher based on FSR is the way to allocate duty to guarantee the safety of the controlling register and the output function. It was discovered that the output function has to have a higher algebraic degree and more inputs to make up for the lower internal state of LWLCM compared to Grain-128. Thus, seven NLFSR bits, one multiplexer bit, one logistic round module bit, and a filter function hide the nonlinearity of 68719476736 ($\approx 2^{36}$). The LWLCM's output function is balanced, outlined over 38 variables, and has the following security features. The primary beginning phase seeks to disorient the contents. Security and speed are traded off in the number of clocks. Should the cipher need frequent initialization using a new IV, initialization efficiency might potentially become a bottleneck. The LFSR includes the IV and 16 ones before starting. If two separate IVs differ by a single bit, the likelihood of the shift register bit remaining the same in both initializations should be approximately 0.5. As explained in Grain v1 [7], this is completed after 160 clocking.

## 5. Proposed work

In this part, we explained the LWLCM encryption and decryption techniques. The proposed cryptography model comprises four essential components: the logistic chaotic module, NLFSR, LFSR, and a multiplexer unit. The amalgamation

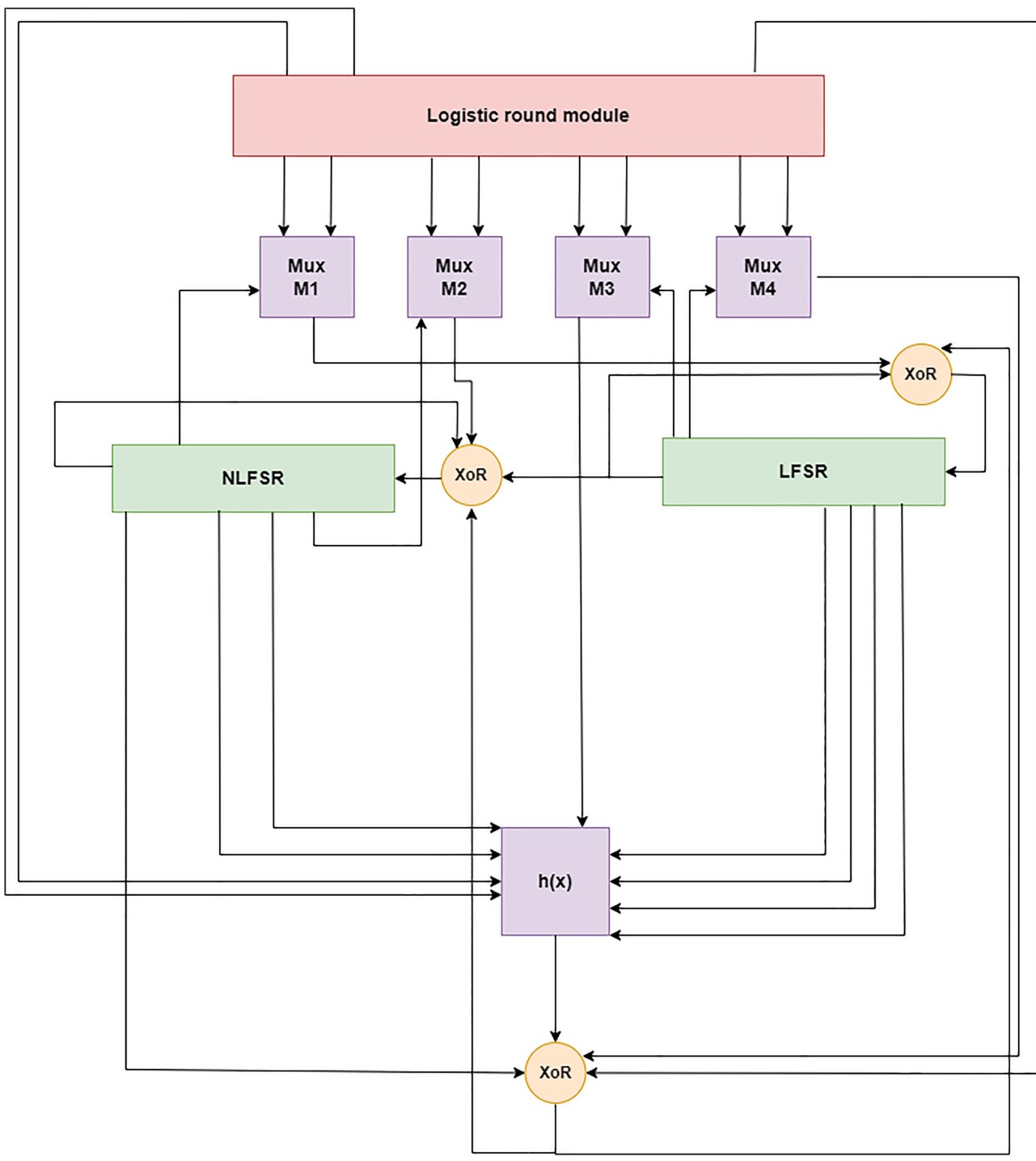

**Fig 5. Key initialization.**

of these elements guarantees the production of exceptionally unpredictable and statistically sound keystreams, which are impervious to cryptanalytic assaults. The core of the randomness production is the logistic chaotic module, which generates a 32-bit pseudorandom sequence derived from the established logistic map, as delineated in Equation 1. This erratic behavior engenders considerable sensitivity to beginning conditions, yielding non-repetitive and decorrelated outputs that augment entropy and diffusion characteristics. The 80-bit LFSR is constructed utilizing a specifically selected primitive polynomial, as indicated in Equation 4, which guarantees the production of a maximal-length sequence with a periodicity of $2^{80} - 1$. The state modification of the LFSR entails designated tap sites and the incorporation of a multiplexer $M_1$. Simultaneously, the NLFSR obtains feedback derived from a nonlinear amalgamation of its own state, the LFSR state, and a secondary multiplexer $M_2$, regulated by the feedback function Equation 6. This design markedly enhances nonlinearity and correlation immunity, hence improving the cipher's resilience against algebraic and linear attacks. The multiplexer unit, essential to both LFSR and NLFSR feedback routes, is a 2 × 1 selector that selects between two input bits according to a single control bit, as formally defined by the feedback function Equation 5. This combination preserves linearity as needed while enabling regulated non-linear perturbations via the multiplexer. This unit presents dynamic switching behavior, which greatly enhances the cipher's confusion feature and guarantees bit-level obfuscation, crucial for mitigating linear and fault-based cryptanalytic methods. To guarantee adequate diffusion and bit intermixing prior to the generation of any keystream output, the system undergoes 160 initial clock cycles without yielding any output. This warm-up step integrates the internal states of LFSR, NLFSR, and the chaotic module, therefore successfully eradicating any patterns resulting from weak initialization vectors or keys. Following this initialization, the 81st clock cycle commences the keystream creation process. The keystream is generated using a composite function that incorporates the feedback mechanisms of LFSR and NLFSR Equations 5 and 7, the nonlinear filter logic Equation 8, and the final output function Equations 11 and 12. The composite output is subsequently XORed with the plaintext bits, producing a safe ciphertext. The suggested design attains an optimal equilibrium between lightweight execution and robust security, rendering it particularly appropriate for resource-limited settings, such as IoT applications. It provides robust randomization properties, little memory usage, and elevated throughput, while ensuring resilience against various cryptanalytic assaults, such as linear, differential, algebraic, and fault attacks. LWLCM's encryption and decryption mechanism is primarily composed of four main components.

1. Using a key and IV, the procedure begins with initializing the LFSR and NFSR. The text_to_bits function converts the IV and key from text to binary. The binary IV sets the first 64 bits of the LFSR, and one is set to the remaining bits. The binary key is used to initialize the NFSR. Set up the NLFSR and LFSR with an 80-bit key and 64-bit IV.

2. A pseudo-random sequence is produced using the logistic map and the parameters $\alpha$ and $x_0$. The iteration function produces the sequence, which regulates how the LFSR and NFSR behave during the encryption/decryption process.

3. The clock function is in charge of updating the status of the LFSR and NFSR registers. Combining bits from the logistic map, LFSR, and NFSR, this function iteratively computes new values for the registers using a sequence of XOR and MUX (multiplexer) operations. The clock function contributes to the unpredictable evolution of the cipher's internal state over time.

The keystream generation function continually generates a keystream using the logistic map, LFSR, and NFSR as a basis. The plaintext and the key stream will be XORed together to create the ciphertext.

### 5.1. Encryption

This section covers the encryption process, the IV, key, and plaintext, and the inputs used in the encryption process after initializing the encryption and "warming it up" with the clock function. The ciphertext is created by XORing the key stream produced by the keystream generation function with the binary plaintext. The binary string is returned in the form of ciphertext as shown in the encryption algorithm.

**Encryption Algorithm**

**Input:** Plaintext (P), α, $x_0$, key (k), Initialization Vector (IV), XOR (⊕)
**Output:** Ciphertext
1. **Derived the initial parameters α, $x_0$ by Equations 2 and 3.**
2. **Initialize the logistic round module:**
   for i = 0 to 32 do
   f[i] = logistic_chaos($x_0$, α) # Compute logistic chaos values by Equation 1.
3. **Initialize NLFSR:**
   for i = 0 to 79 do
     nlfsr[i] = k[i] # Load key values into NLFSR
4. **Initialize LFSR with IV:**
   for i = 0 to 63 do
   fsr[i] = IV[i] # Load initialization vector values into LFSR
5. **Continue initializing LFSR:**
   for i = 64 to 79 do
   lfsr[i] = 1 # Set LFSR values from 64 to 79 to
6. **Clock the cipher 160 times without generating a keystream:**
   for j = 0 to 159 do
   clock() # Apply the clocking function using Equations 5, 6, and 11
7. **Generate keystream:**
   keystream=keyStreamGeneration() # Apply the clocking function using Equations 5, 6, 11, and 12
8. **Encrypt plaintext:**
   for i = 0 to length(P) - 1 do
   Ciphertext[i] = P[i] ⊕ keystream[i] # XOR plaintext with keystream
9. **Return Ciphertext**

## 5.2. Decryption

Encryption and decryption are the opposite processes. The decryption function initializes with the same IV and key, executes the clock function, and generates the same key stream. The binary ciphertext and key stream are XORed to return the original binary plaintext. This binary plaintext is then converted back into text using text_from_bits, resulting in the original message shown by decryption algorithms.

**Decryption Algorithm**

**Input:** Cipher (C), α, $x_0$, key (k), Initialization vector (IV), XOR (⊕)
**Output:** Plaintext
1. **Derived the initial parameters α, $x_0$ by Equations 2 and 3.**
2. **Initialize logistic chaos module**
   for i=0 to 32 do
   f[i] = logistic_chaos($x_0$, α) # Compute logistic chaos values by Equation 1.
3. **Initialize NLFSR**
   for i=0 to 79 do
   nlfsr[i] = k[i]
4. **Initialize LFSR with IV**
   for i=0 to 63 do
   lfsr[i] = IV[i]
5. **Further initialize LFSR**
   for i=64 to 79 do
   lfsr[i] = 1
6. **Clock the 160 times #using Equations 5, 6, and 11 ciphers without generating a keystream**
   clock()
7. **Generate the keystream # using Equations 5, 6, 11, and 12**
   keystream=keyStreamGeneration()

```
8. Decrypt the ciphertext
   for i = 0 to length(Cipher) do
   Plaintext[i] = Cipher[i] ⊕ keystream[i]
9. Return Plaintext
```

## 6. Results and discussion

The proposed model was evaluated using Python on a Raspberry Pi 4 platform to simulate a genuine IoT environment, as shown in Figs 6 and 7. This study employed synthetic text files from the 20 Newsgroups dataset from Kaggle. Over 20,000 newsgroup posts exist across 20 distinct groups of varying sizes, such as $2^6$, $2^8$, $2^{10}$, and $2^{12}$ bytes, as plaintexts to evaluate performance. The files include alphanumeric characters, akin to the data transmitted by IoT devices, such as sensor readings, command packets, or configuration records. We select these plaintexts to demonstrate the method's efficacy with various types and amounts of data required for lightweight IoT connectivity. All encrypted files must be stored in UTF-8 format to facilitate byte-level bitstream conversion. To ensure the evaluation was equitable and uniform, we contrasted the proposed LWLCM stream cipher with established lightweight and classical algorithms, including AES-256, Grain v1, Grain-128, Trivium, and RSA-1024. We employed a replication method to compare the algorithms. This necessitated writing the code for each algorithm independently. Subsequently, we employed the identical gear and software for all of them. All tests for encryption and decryption utilized the same plaintext files. This was executed to ensure uniformity of all criteria. To eliminate bias and ensure the outcomes were as consistent as possible, the results were averaged over 100 iterations for each data size. Python's timing utilities were used to evaluate the length of encryption and energy

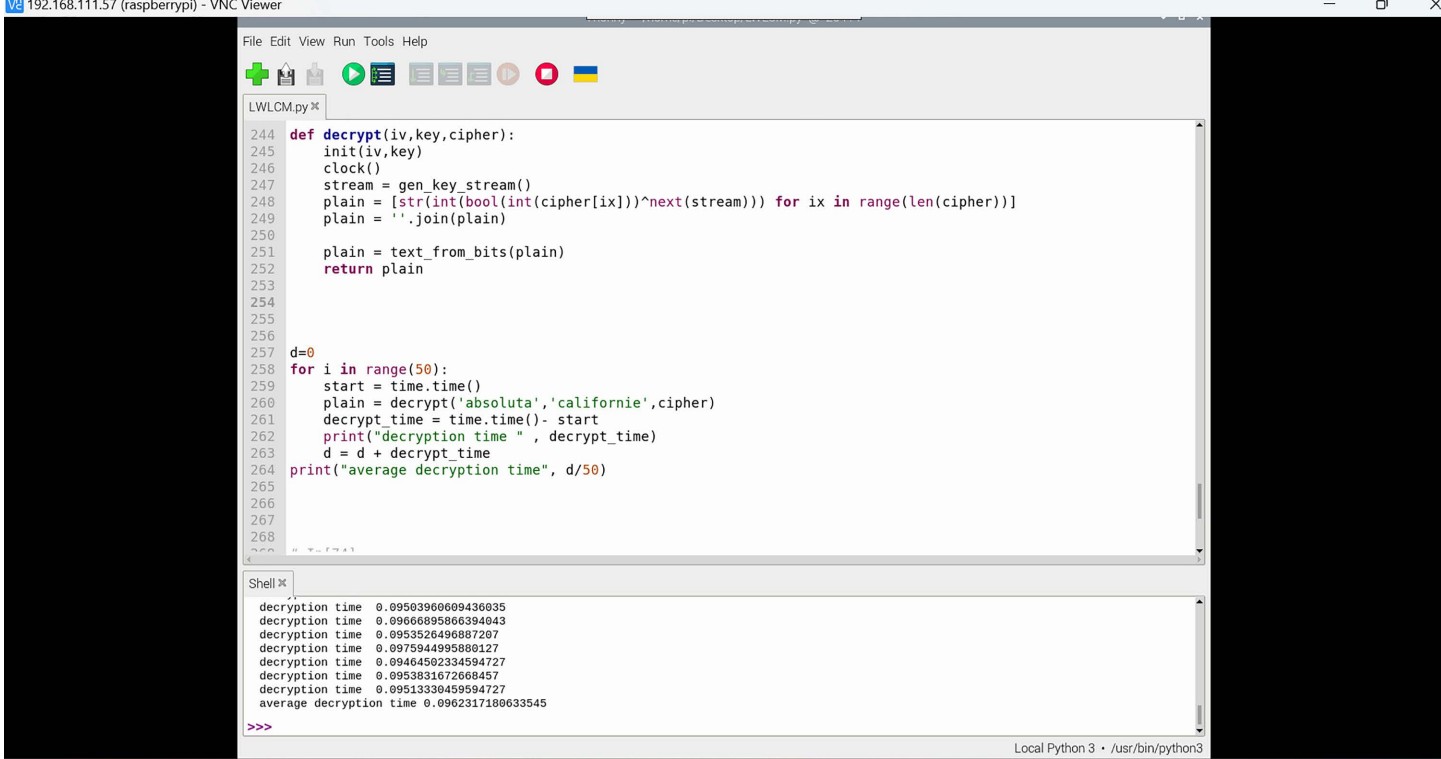

**Fig 6. Simulation of the LWLCM on Raspberry Pi.**

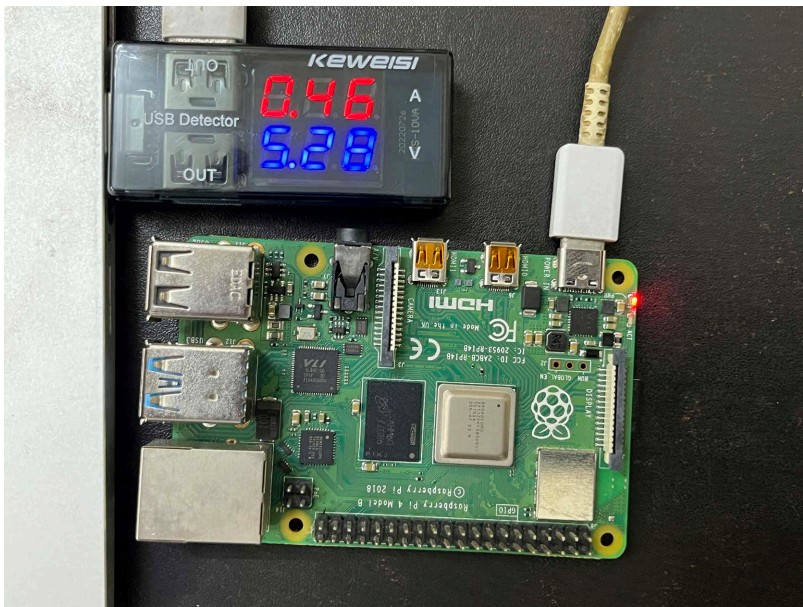

**Fig 7. Experimental setup of Raspberry Pi.**

consumption. Current was calculated using current sensors, while an average device voltage pull of 5V was used to guide energy computation. The tracemalloc module was utilized to monitor memory usage. NumPy was used for efficient numerical computation, while Pandas was employed for data organizing, averaging, and result aggregation. To ensure resilience and unpredictability, we assessed the encryption by Shannon entropy, avalanche effect, correlation coefficient, and the comprehensive NIST randomness test suite. Time complexity analysis is done to check the performance of the proposed model on IoT devices. Additionally, matplotlib was utilized to provide visual comparisons and charts depicting encryption time curves, entropy distributions, and avalanche variation graphs for different plaintext sizes, alongside typical ciphers including AES-256, Grain-128, Grain v1, Trivium, DES, and RSA. These graphical representations demonstrate the proposed encryption's performance advantages and enhance the clarity of the testing results. Further ablation experiments were conducted to evaluate the impact of each component on the quality of the keystream and the execution speed. This illustrates the significance of each component and the impact of them on crucial parameters like encryption/decryption time, throughput, avalanche effect, and entropy. The metrics that we have evaluated to check the performance of the proposed model and how these metrics are related to a real-world IoT scenario are discussed below:

## 6.1. Time complexity analysis

With n representing the bit count in the input plaintext or ciphertext, the proposed lightweight stream cipher defines its time complexity as $O(n)$. The encryption process consists of five critical stages: first, the plaintext is converted into binary format, requiring $O(n)$ time; second, the LFSR and NLFSR are initialized with the designated key and initialization vector (IV) in constant time, $O(1)$; third, a warm-up phase of 160 fixed rounds is performed to diffuse the internal state, also in $O(1)$ time; fourth, the keystream is generated with each bit calculated in constant time, resulting in an overall complexity of $O(n)$; and finally, each bit of the plaintext is XORed with the corresponding keystream bit during encryption, which again necessitates $O(n)$ time. The overall time complexity of the encryption and decryption processes is $T(n) = O(n) + O(1) + O(1) + O(n) + O(n) \approx O(n)$, which indicates that the method exhibits linear scalability for input size, making it particularly appropriate for resource-constrained environments such as IoT devices.

## 6.2. Encryption time

In IoT and real-time applications, encryption time is a critical metric of cryptographic efficiency, defined as the duration needed for an encryption method to transform plaintext into ciphertext. Further, based on the size of each data set, four sets of data are gathered, and the encryption time is evaluated to clarify whether the proposed model is suitable for IoT-based applications. To find the encryption time, 100 trials are performed for each input type. The average values of the final results have been considered and are shown in Table 2, and a comparison has been made in Fig 8 indicates that the proposed LWLCM cipher operates at a reduced speed compared to AES-256, Grain v1, and Trivium, mostly attributable to the incorporation of a nonlinear chaotic function and dynamic multiplexer logic within its architecture. By augmenting the internal complexity and computational depth of the methodology, these components enhance security attributes in the keystream, including statistical unpredictability, nonlinearity, and increased diffusion. This additional complexity is a calculated compromise to provide enhanced resilience to linear, differential, and algebraic attacks, hence augmenting the security of LWLCM in adversarial conditions. For all plaintext values, LWLCM demonstrates an encryption speed that is around 56–59% quicker than Grain-128 and nearly 94–95% faster than RSA. Grain-128 experiences prolonged processing times due to its enlarged state size and intricate feedback mechanisms, notwithstanding its security; RSA is fundamentally unsuitable for bulk data encryption because of its asymmetric nature and computational demands. Consequently, LWLCM outperforms Grain-128 and RSA in execution time while offering superior cryptographic protection compared to faster yet less sophisticated ciphers such as Grain v1 and Trivium, thereby optimizing lightweight efficiency with better security.

**Table 2. Encryption time (sec).**

| Plain text length (Byte) | AES-256 | Grain v1 [7] | Grain-128 [8] | Trivium [5] | LWLCM | RSA |
|---|---|---|---|---|---|---|
| $2^6$ | 0.08 | 0.07541 | 0.2318770 | 0.04522 | **0.09957** | 1.5 |
| $2^8$ | 0.25 | 0.2831 | 0.74148 | 0.12840 | **0.3100** | 5.7 |
| $2^{10}$ | 0.90 | 1.09217 | 2.7381 | 0.45940 | **1.20560** | 22.3 |
| $2^{12}$ | 3.65 | 4.3283 | 11.5931 | 1.8176 | **4.7270** | 90 |

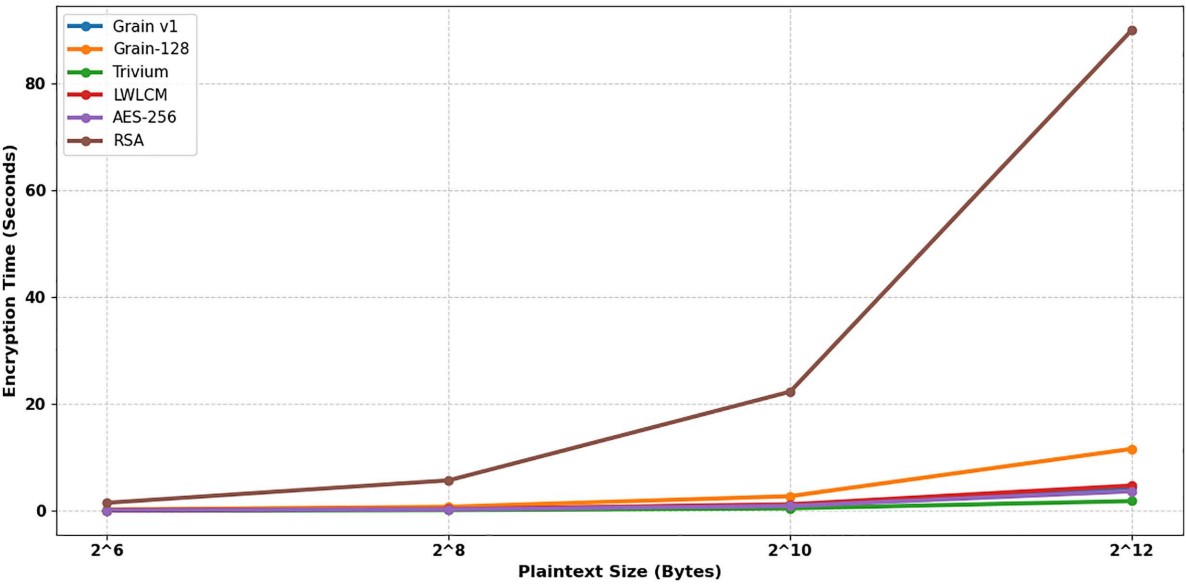

**Fig 8. Encryption time (sec).**

## 6.3. Decryption time

The decryption process in the proposed model utilizes a stream cipher technique, in which the ciphertext is XORed with the keystream to obtain the original plaintext. Table 3 and Fig 9. The proposed LWLCM cipher consistently outperforms both Grain-128 and RSA in terms of decryption performance across all evaluated plaintext sizes. Specifically, LWLCM demonstrates a 56% to 59% reduction in decryption time compared to Grain-128, owing to its lightweight structure and efficient feedback mechanisms. When compared to RSA, the performance advantage is even more significant—LWLCM achieves a 75% to 85% faster decryption time, highlighting its suitability for resource-constrained environments where computational efficiency and low latency are critical. While the proposed work is marginally slower than Grain v1, especially for smaller plaintext sizes (10–14% slower), the difference disappears for bigger plaintexts and is less than 0.5 percent slower because of its complex nature. The proposed work strikes a compromise between security and efficiency, providing notable advancements over Grain-128 and RSA while maintaining parity with Grain v1, Trivium, and AES-256.

## 6.4. Throughput

Throughput in cryptography refers to the rate at which an encryption or decryption algorithm processes data. It determines how efficiently an algorithm can handle large volumes of data in real-time applications. The throughput of the algorithm grows as its performance grows because there is a direct correlation between the two [32]. The encryption/decryption method's throughput can be ascertained using Equations 13 and 14.

**Table 3. Decryption Time(sec).**

| Plain text length (Byte) | Grain v1 [7] | Grain-128 [8] | Trivium [5] | LWLCM | AES-256 | RSA |
|---|---|---|---|---|---|---|
| $2^6$ | 0.08446 | 0.2371 | 0.04582 | 0.09623 | 0.07 | 0.13 |
| $2^8$ | 0.2831 | 0.7398 | 0.12855 | 0.31230 | 0.23 | 0.52 |
| $2^{10}$ | 1.09217 | 2.7381 | 0.46390 | 1.20380 | 0.86 | 2.03 |
| $2^{12}$ | 4.32830 | 11.474 | 1.79119 | 4.72630 | 3.59 | 8.27 |

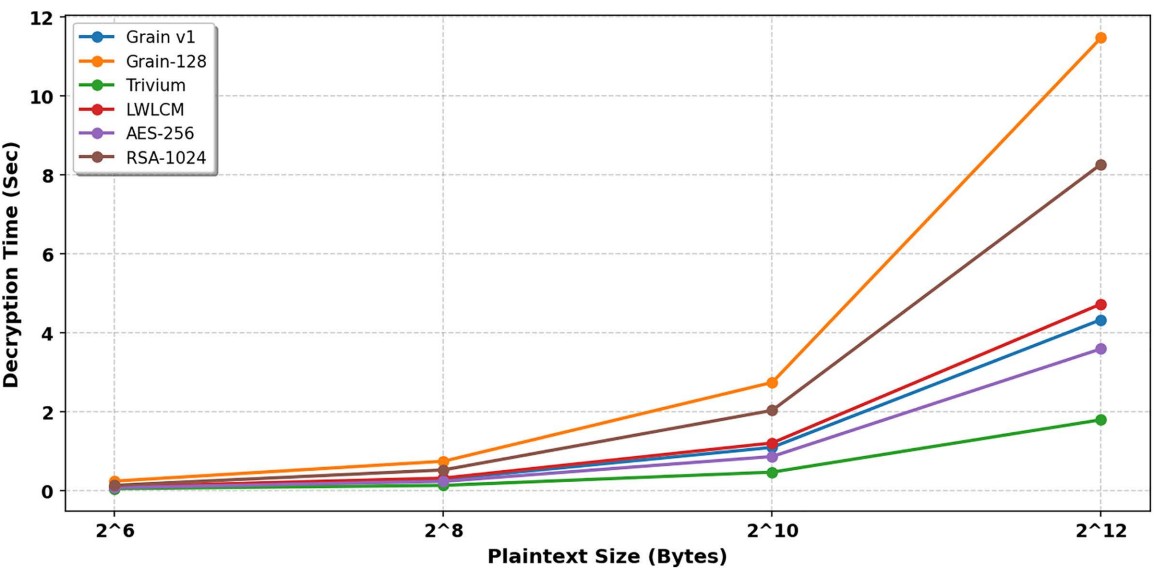

**Fig 9. Decryption time (sec).**

$$Encryption_{Throughput} = \frac{Encryption_{time}}{plaintext_{lenght}}$$

(13)

$$Decryption_{Throughput} = \frac{Decryption_{time}}{Ciphertext_{length}}$$

(14)

Tables 4,5, and Figs 10 and 11 show that the LWLCM cipher consistently demonstrates superior encryption and decryption throughput compared to Grain-128 and RSA-1024 across all assessed plaintext sizes. LWLCM demonstrates a throughput enhancement of around 130% to 150%, surpassing Grain-128 and underscoring its design focused on lightweight efficiency. The throughput enhancement of 1800% is particularly significant when juxtaposed with RSA-1024, hence emphasizing the efficacy of LWLCM in low-resource settings. While Trivium attains the maximum raw encryption/decryption throughput, followed by AES-256 and Grain v1. LWLCM presents a judicious compromise among speed, randomness, and cryptographic strength due to its nonlinear chaotic value and multiplexer unit, with a tradeoff between security and efficiency. LWLCM performs admirably compared to Grain-128 and RSA-1024, providing a notably higher throughput at a reasonable cost compared to Grain v1.

## 6.5. Avalanche analysis

In cryptography, the term "avalanche effect" is used to characterize a specific behavior of mathematical functions used in encryption. The avalanche effect is one of the desirable properties of any encryption method. A single-bit change in the key should result in a 50% change in the ciphertext or keystream. This characteristic is known as the "avalanche effect." [33]. This quality is crucial in preventing opponents from using plaintext attacks or statistical analysis to take advantage of weaknesses in encryption. In the proposed algorithm, a single-bit flip was introduced at any $i^{th}$ bit of the secret key, keeping the IV constant. This update changed how the NLFSR was set up. We looked at the ciphertext that came out and the original ciphertext to determine how many bits had changed. Examining LWLCM on both pairs of key-ciphertext and key-keystream, as shown in Tables 6 and 7, Figs 12 and 13, exposes a more noticeable avalanche impact than modern lightweight stream ciphers such as Grain v1, Trivium, and AES-256. The results show that above 50%, LWLCM consistently achieves an avalanche effect over the required security threshold on both pairs. The great unpredictability of the cipher strengthens its resilience by ensuring that small changes in input produce significant differences in ciphertext or

Table 4. Encryption throughput (Byte/sec).

| Plain text length (Byte) | Grain v1 [7] | Grain-128 [8] | Trivium [5] | LWLCM | AES-256 | RSA-1024 |
|---|---|---|---|---|---|---|
| $2^6$ | 848 | 276 | 1415 | **642** | 800.0 | 43 |
| $2^8$ | 904 | 345 | 1993 | **825** | 1024 | 45 |
| $2^{10}$ | 937 | 373 | 2228 | **849** | 1137 | 46 |
| $2^{12}$ | 946 | 353 | 2253 | **866** | 1122 | 45 |

Table 5. Decryption throughput (Byte/sec).

| Plain text length (Byte) | Grain v1 [7] | Grain-128 [8] | Trivium [5] | LWLCM | AES-256 | RSA-1024 |
|---|---|---|---|---|---|---|
| $2^6$ | 757 | 269 | 1396 | **665** | 914 | 492 |
| $2^8$ | 904 | 346 | 1991 | **819** | 1113 | 492 |
| $2^{10}$ | 937 | 373 | 2207 | **850** | 1190 | 504 |
| $2^{12}$ | 946 | 356 | 2286 | **866** | 1140 | 495 |

Fig 10. Encryption throughput (Kb/sec).

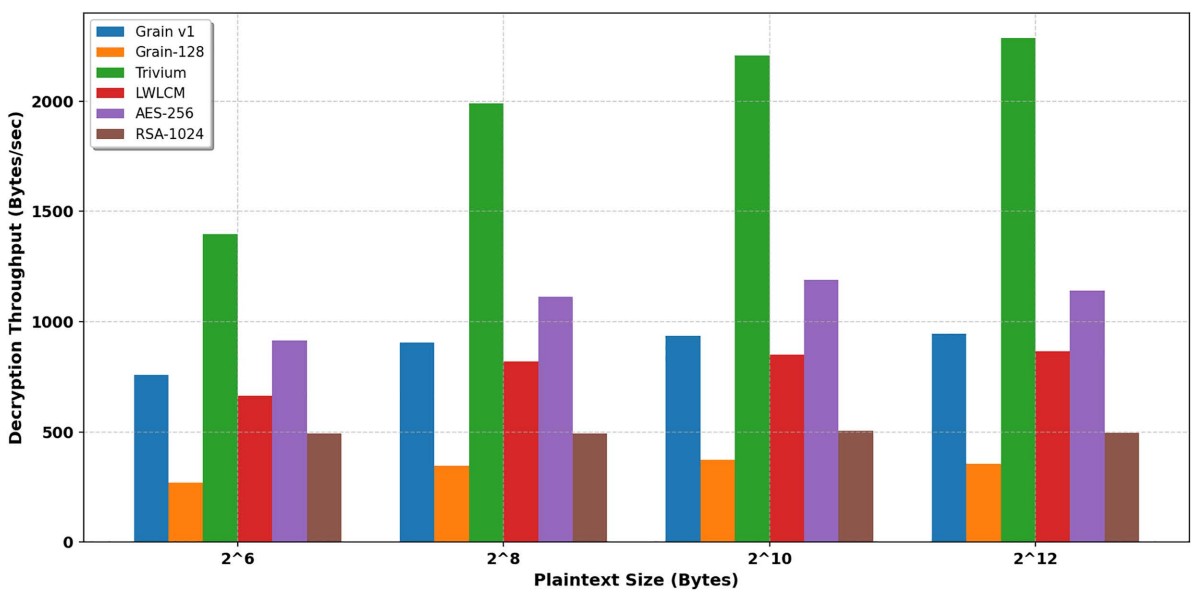

Fig 11. Decryption throughput (kb/sec).

Table 6. Avalanche Effect (%) of Key- Keystream.

| Plain text length (Byte) | Grain v1 [7] | Grain-128 [8] | Trivium [5] | LWLCM | AES-256 | RSA-1024 |
|---|---|---|---|---|---|---|
| $2^6$ | 52.92 | 53.3203 | 49.9804 | **52.53** | 50.12 | 57.2 |
| $2^8$ | 49.88 | 49.942 | 49.8256 | **49.95** | 49.95 | 56.4 |
| $2^{10}$ | 49.78 | 50.5737 | 49.914 | **50.17** | 50.08 | 54.9 |
| $2^{12}$ | 49.94 | 50.238 | 50.033 | **50.53** | 50.19 | 55.8 |

**Table 7. Avalanche effect (%) Key- Cipher text.**

| Plain text length (Byte) | Grain v1 [7] | Grain-128 [8] | Trivium [5] | LWLCM | AES-256 | RSA-1024 |
|---|---|---|---|---|---|---|
| $2^6$ | 52.92 | 53.3203 | 49.9804 | **50.7812** | 50.18 | 57.5 |
| $2^8$ | 49.78 | 50.5737 | 49.9145 | **50.0488** | 49.99 | 56.7 |
| $2^{10}$ | 49.94 | 50.238 | 50.0335 | **49.3743** | 50.04 | 55.2 |
| $2^{12}$ | 49.88 | 49.9442 | 49.8256 | **49.9473** | 50.21 | 55.9 |

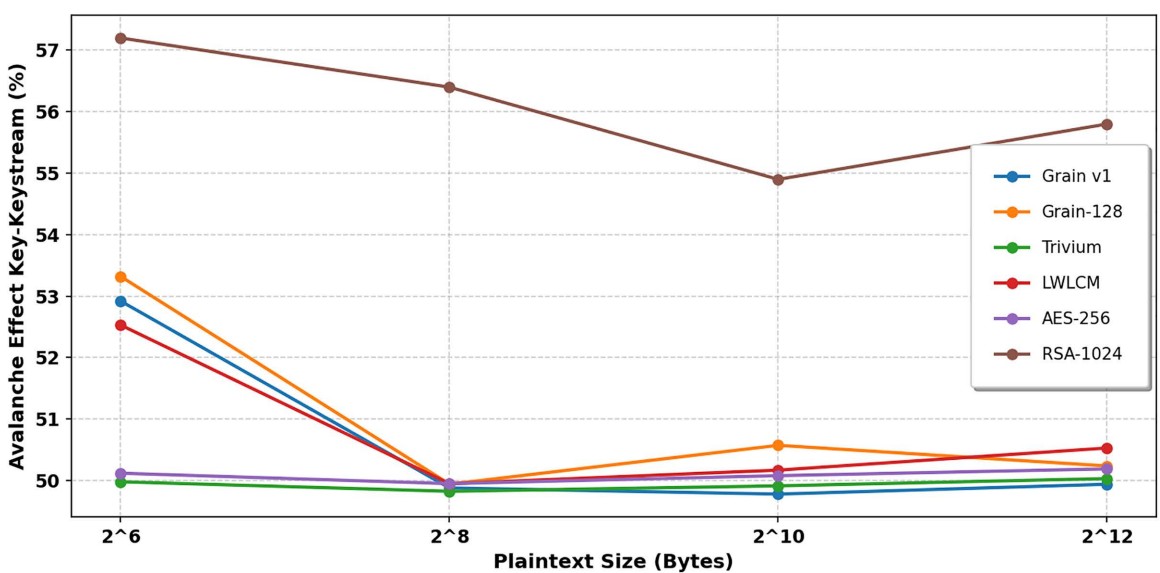

**Fig 12. Key-Keystream Avalanche effect.**

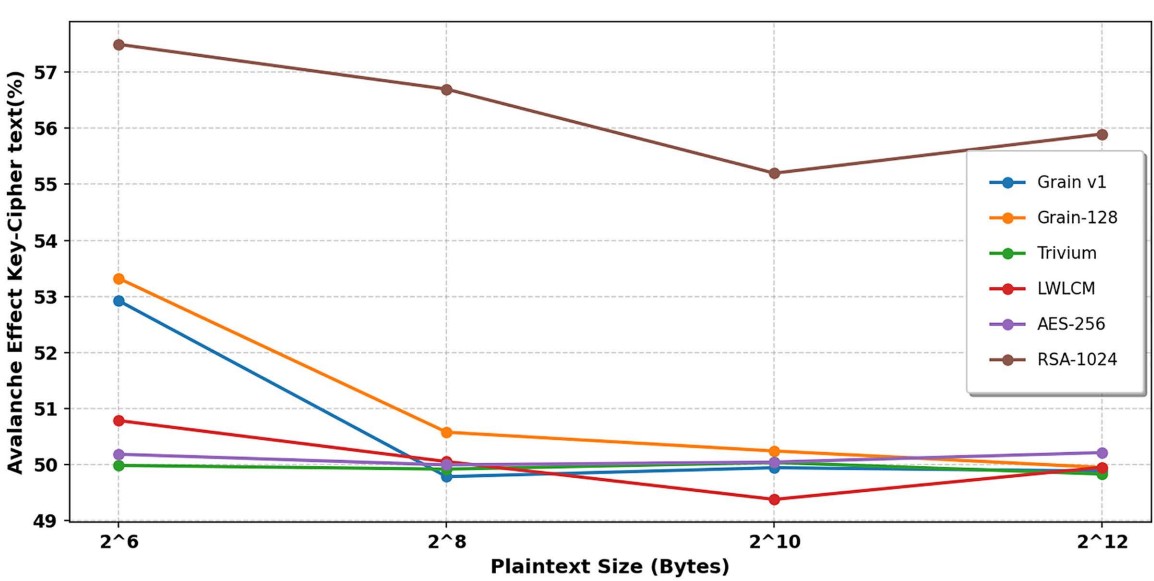

**Fig 13. Key-Cipher Text Avalanche effect.**

keystream, complicating cryptanalysis greatly. The encryption model integrates the logistic chaos function to increase security by adding non-linearity and enhanced sensitivity to initial conditions. Including the chaotic function ensures that even small changes in the input generate drastically different keystreams, hence improving diffusion characteristics and preventing attackers from recognizing regular patterns. Figs 14 and 15 show that adding logistic chaos to LWLCM produces an improved average avalanche effect on both pairs of key-ciphertext and key-keystream and outperforms Grain v1, Trivium, and AES-256, increasing the cipher's resistance against cryptographic attacks. The results conclude that the proposed approach significantly increases security while preserving efficiency, offering resilience to statistical and differential attacks. Its unpredictability, strong diffusion, and low computing cost make it a perfect solution.

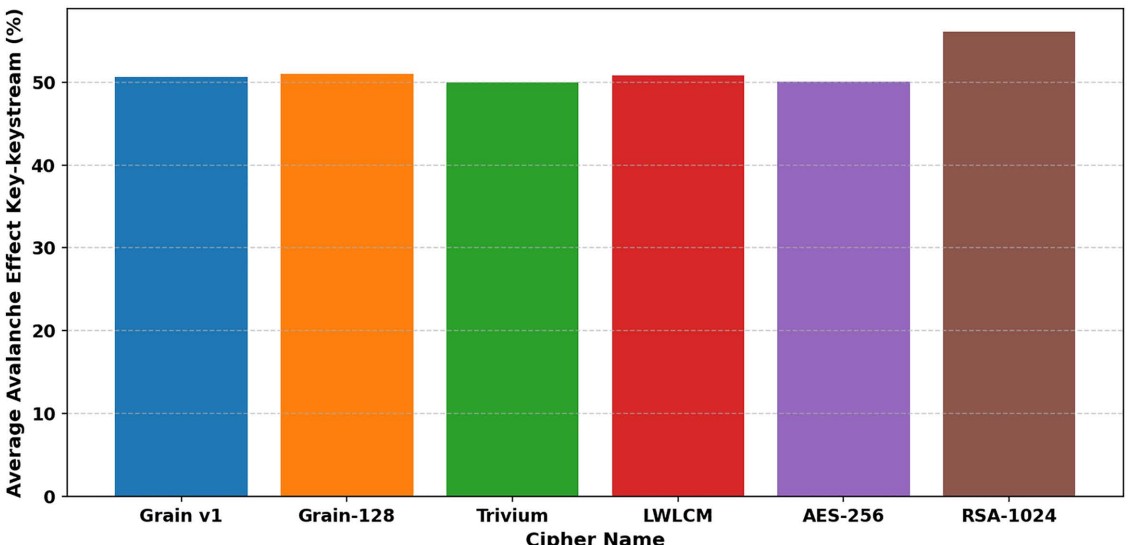

**Fig 14. Average Avalanche effect for Key-Keystream.**

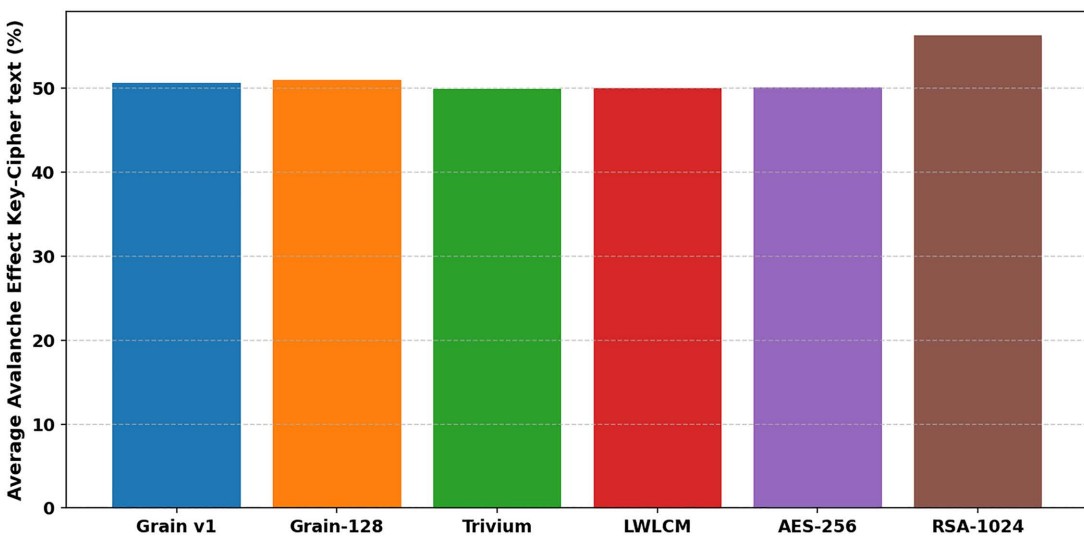

**Fig 15. Average Avalanche effect for Key-Cipher text.**

## 6.6. Memory usage

In the domain of IoT, where devices operate with constrained memory, power, and computational resources, efficient memory management is a critical factor in evaluating the feasibility of cryptographic algorithms. Any algorithm's lowest and maximum memory usage, quantified in kilobytes (KB), directly affects its suitability for real-time IoT applications. Fig 16 depicts the variance in memory utilization among LWLCM, Grain v1, Grain-128, and Trivium, highlighting the compromise between security and resource efficiency. The integration of a multiplexer and logistic chaos map intensifies code complexity, leading to heightened memory usage. However, this heightened complexity significantly enhances security, making the proposed method more resilient than traditional lightweight algorithms. The balance between memory optimization and security robustness is essential for developing scalable, energy-efficient, and attack-resistant cryptographic solutions in IoT environments.

## 6.7. Energy consumption

Energy consumption is essential in assessing how well algorithms operate in limited settings. The efficiency with which the algorithm functions with constrained resources like battery-operated devices is amply demonstrated by this [34]. We can evaluate how well the suggested approach balances security and cost by monitoring energy usage. We can calculate energy consumption by using Equation 15.

$$Energy_{consumption} = V \times I \times t \tag{15}$$

V is the potential difference, I is the current, and t is the total time. The LWLCM's energy usage demonstrates how well it balances efficiency and improved security measures, as shown in Table 8 and Fig 17. It uses more energy than Grain v1, Trivium, and AES-256 due to its complex nature. However, this difference is offset by the more sophisticated security features that provide better defense against intrusions. All plaintext sizes combined show that LWLCM is still substantially

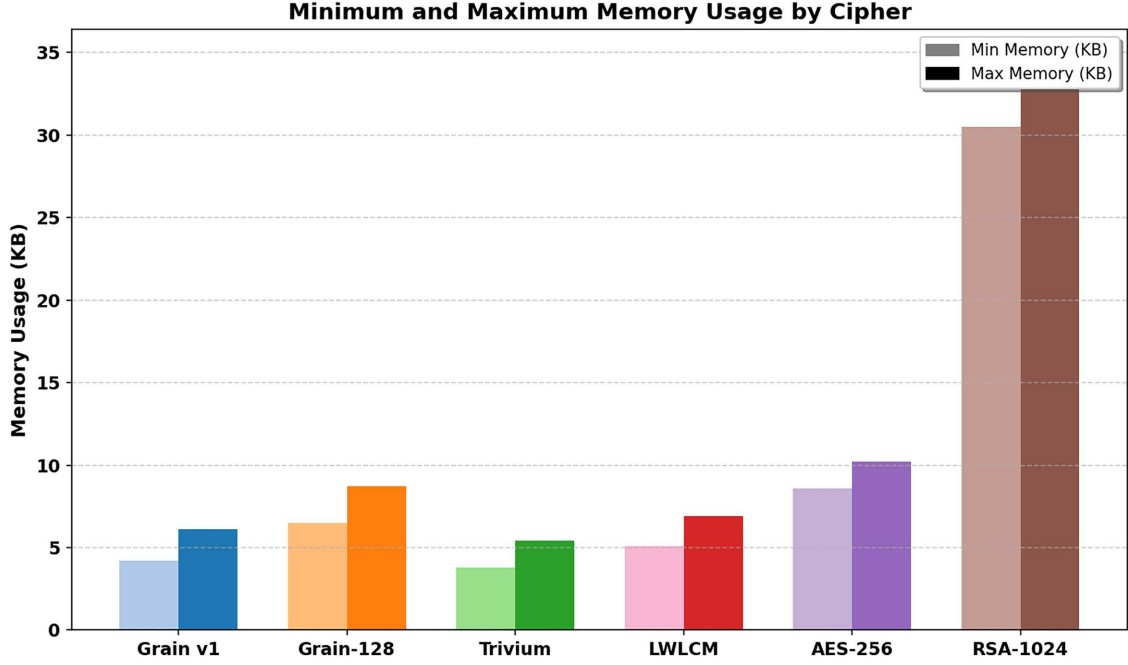

**Fig 16. Memory usage (Kb).**

**Table 8. Energy consumption (Joule).**

| Plain text length (Byte) | Grain v1 [7] | Grain-128 [8] | Trivium [5] | LWLCM | AES-256 | RSA-1024 |
|---|---|---|---|---|---|---|
| $2^6$ | 0.237 | 0.730413 | 0.14696 | **0.313646** | 0.2 | 3.75 |
| $2^8$ | 0.934 | 2.298588 | 0.40125 | **1.023** | 0.625 | 14.25 |
| $2^{10}$ | 3.549 | 8.351205 | 1.47008 | **4.09904** | 2.25 | 55.75 |
| $2^{12}$ | 13.85 | 35.93861 | 5.9072 | **15.1264** | 9.125 | 225 |

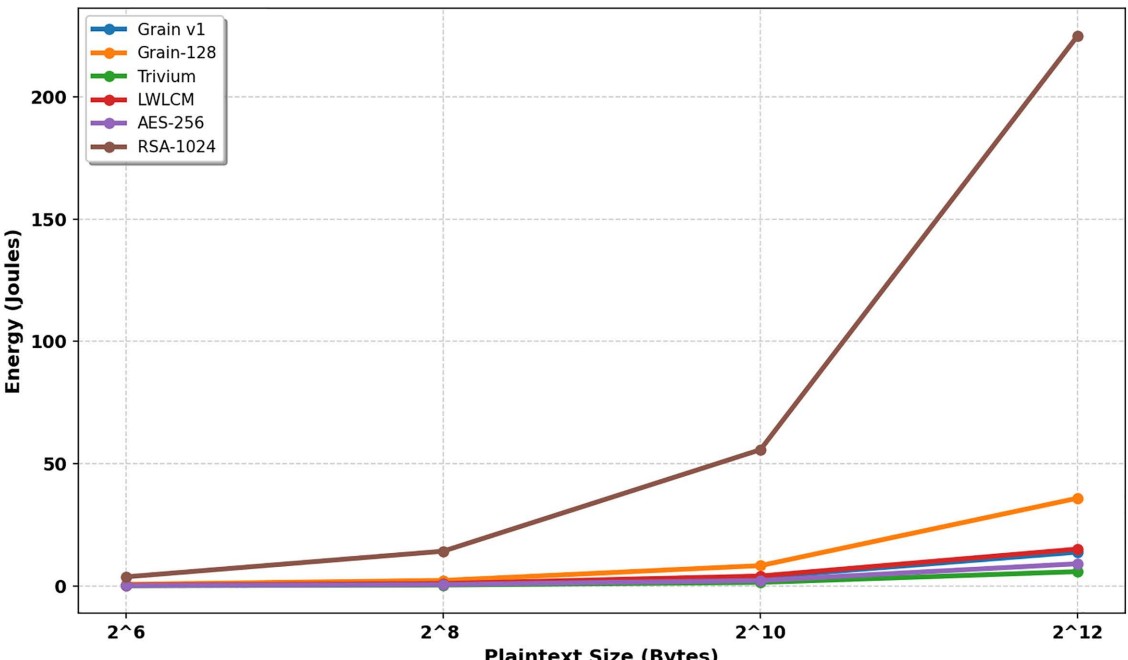

**Fig 17. Energy consumption (Joule).**

more energy-efficient than Grain-128 and RSA-1024. Because of this, LWLCM is an efficient cipher that provides strong security without requiring as much energy as more complicated ciphers like Grain-128 and RSA-1024. Its effectiveness in situations where security and efficiency are critical for solid encryption procedures is highlighted by its capacity to deliver robust digital privacy while consuming reasonable energy.

Analyzing the proposed cipher's efficacy reveals an essential equilibrium between security and velocity. Despite prioritizing lightweight operations, which may sacrifice some cryptographic complexity, ciphers such as Trivium, Grain v1, and AES-256 demonstrate enhanced encryption and decryption speeds alongside improved throughput. This trade-off enhances rapid data processing but may slightly diminish advanced randomization or vulnerability to certain attacks. The proposed LWLCM encryption exhibits greater structural complexity through the utilization of a logistic chaos function and dynamic multiplexer logic. Statistical analyses indicate that these components markedly enhance the overall unpredictability, diffusion effectiveness, and resilience against cryptanalytic assaults of the cipher, despite a slight prolongation of the encryption duration. The proposed LWLCM cipher significantly surpasses Grain-128 and RSA-1024 in terms of encryption speed during benchmarking. The extended processing times of Grain-128 and RSA-1024 are attributable to their elevated state dimensions and inherently complex computations. While LWLCM enhances security layers, it preserves a superior speed compared to conventional approaches, therefore affirming its efficiency. Ultimately, LWLCM attains an optimal

equilibrium: it is marginally slower than Trivium, Grain v1, and AES-256, while significantly faster than Grain-128 and RSA, hence enhancing cryptographic security. This equilibrium renders LWLCM an advantageous option for restricted environments such as IoT for immediate, secure communication.

## 6.8. Ablation study on component impact

We performed an ablation study to assess the influence of each element of the LWLCM cipher on the efficacy of the data encryption process. We gradually removed the NLFSR, the chaotic module, and the MUX, yielding three simplified ciphers. We ensured equitable outcomes by utilizing uniform sizes of plaintext and settings for each configuration test. From Table 9, the results show that removing any of the modules speeds up the encryption process, notably when the chaotic function is not present. In this case, it takes 4.7270 seconds to encrypt $2^{12}$ bytes of plaintext with the complete LWLCM and 2.242 seconds without the chaos module. The same patterns still show up if we remove the MUX (0.860 s) or the NLFSR (0.863 s). This shows that these modules make arithmetic a lot harder. Now, in the case of the decryption process, the same thing has been observed from Table 10. On removing each individual module, the decryption time decreases.

Further, the throughput has also been calculated to check the efficiency of each module. Table 11 demonstrates that the LWLCM model may encrypt data using four alternative approaches. The results demonstrate that dropping the chaotic module markedly optimizes throughput. The thorough model can process 4096 bytes at a velocity of 866.71 bytes per second. The new model is capable of processing 1826.74 bytes per second. The performance improves slightly when the MUX and NLFSR are removed. Certain electrical equipment operates more efficiently with increased data transmission; yet, this results in diminished randomness and linearity of the data. Incorporating chaos and multiplexing into the encryption process enhances complexity, hence complicating decryption and facilitating communication; however, it demands substantial computational resources. This indicates their necessity for the final design.

**Table 9. Encryption time(sec) in the absence of each component at a time.**

| Plain text length(byte) | Without chaos | Without Mux | Without NLFSR | LWLCM |
|---|---|---|---|---|
| $2^6$ | 0.0661 | 0.014 | 0.031 | 0.09957 |
| $2^8$ | 0.090 | 0.048 | 0.063 | 0.3100 |
| $2^{10}$ | 0.392 | 0.253 | 0.286 | 1.20560 |
| $2^{12}$ | 2.242 | 0.860 | 0.863 | 4.7270 |

**Table 10. Decryption time(sec) in the absence of each component at a time.**

| Plain text length(byte) | Without chaos | Without Mux | Without NLFSR | LWLCM |
|---|---|---|---|---|
| $2^6$ | 0.0656 | 0.012 | 0.028 | 0.09623 |
| $2^8$ | 0.087 | 0.043 | 0.061 | 0.31230 |
| $2^{10}$ | 0.390 | 0.251 | 0.282 | 1.20380 |
| $2^{12}$ | 2.235 | 0.857 | 0.861 | 4.72630 |

**Table 11. Throughput (byte/sec) in the absence of each component at a time.**

| Plaintext Size (bytes) | Without Chaos | Without MUX | Without NLFSR | LWLCM |
|---|---|---|---|---|
| $2^6$ | 393.34 | 1857.14 | 838.71 | 261.13 |
| $2^8$ | 311.11 | 583.33 | 444.44 | 90.32 |
| $2^{10}$ | 2612.24 | 4046.64 | 3578.32 | 849.49 |
| $2^{12}$ | 1826.74 | 4762.79 | 4745.65 | 866.71 |

Other important parameters that evaluate the randomness of the keystream are entropy and the Avalanche effect to protect against different attacks. Table 12 presents the entropy of ciphertexts generated by the LWLCM cipher with various structural features deactivated. The fundamental concept is that the combined utilization of chaotic maps, MUX, and NLFSR consistently approaches the optimal entropy level of 8.0. This renders forecasting future events exceedingly challenging. Upon the removal of the chaotic module, the entropy values significantly diminish (for instance, 7.64 for a $2^8$-byte input), indicating a reduction in randomness. Removing the NLFSR also results in a decrease, albeit less significant. Removing the MUX somewhat increases the entropy; however, it remains marginally lower than that of the entire system. These results indicate that each module is crucial for ensuring the statistical security of the encryption. The chaotic function is crucial as it enhances bit-level diffusion and non-linearity. While in the case of the avalanche effect, the proposed LWLCM encryption consistently exhibits an avalanche effect nearing the optimal 50% threshold across all tested plaintext sizes. From Table 13, it can be observed that upon removal of the chaotic module, the impact of the avalanche diminishes by around 5–6%. This indicates that chaos is a crucial element in rendering the ciphertext less discernible and more dispersed. This means that chaos is an important part of making the ciphertext less clear and more spread out. When the avalanche's performance drops to about 41–42% without the MUX, the difference becomes clearer. This shows that the multiplexer is necessary to make the encryption output less linear. Removing the NLFSR reduces the avalanche effect by around 3–4%. This complicates the interpretation of the input and obscures the clarity of the findings. These results demonstrate the efficacy of LWLCM's architecture and the significance of each component in achieving robust statistical security. The integrated architecture ensures that even a minor alteration to the key will result in a significant, unforeseen change in the ciphertext. This complicates the ability of cryptographers to obtain the keys necessary for message decryption.

## 7. Security Analysis

The following section examines widespread stream cipher attacks and their applicability to LWLCM. A new cipher's ability to withstand all known cryptanalytic assaults is its most important feature [35]. We examine the suggested encryption's resilience against prevalent attacks such as linear cryptanalysis, algebraic attacks, and fault attacks to assess its cryptographic robustness. The primary defense mechanism is evident in the significant nonlinearity generated by multiple sources.

- **Logistic chaos module:** Produces erratic sequences characterized by elevated entropy.

- **NLFSR design:** Incorporates nonlinear feedback functions featuring cross terms.

- **MUX logic:** Incorporates input-dependent control switching, thus obscuring the correlation between input and output.

**Table 12. Entropy analysis in the absence of each component.**

| Plaintext Size (bytes) | Without Chaos | Without MUX | Without NLFSR | LWLCM |
|---|---|---|---|---|
| $2^6$ | 7.68 | 7.82 | 7.77 | 7.999952 |
| $2^8$ | 7.64 | 7.85 | 7.79 | 7.999984 |
| $2^{10}$ | 7.70 | 7.88 | 7.82 | 7.998584 |
| $2^{12}$ | 7.73 | 7.90 | 7.84 | 7.999888 |

**Table 13. Avalnche analysis(%) in the absence of each components at a time.**

| Plaintext Size (bytes) | LWLCM | Without Chaos | Without MUX | Without NLFSR |
|---|---|---|---|---|
| $2^6$ | 50.7812 | 48.021 | 46.7983 | 45.4482 |
| $2^8$ | 50.0488 | 47.9422 | 46.1139 | 44.8653 |
| $2^{10}$ | 49.3743 | 46.8754 | 45.6721 | 44.1175 |
| $2^{12}$ | 49.9473 | 46.9882 | 45.7594 | 44.9983 |

The output function dynamically integrates several register states and chaotic bits by a combination of AND, XOR, and MUX operations. This structure renders it infeasible to represent the outcome using a low-degree algebraic expression or to approximate it with basic linear methods. The estimated nonlinearity of around $2^{38}$ is derived from the quantity of nonlinear monomial terms and the combinational logic depth in the keystream generation process, surpassing that of numerous traditional stream ciphers (e.g., Trivium $\sim 2^{32}$). Despite the dynamic and chaotic nature of the Boolean function rendering precise computation of its Algebraic Normal Form(ANF)computationally challenging, we corroborate this assessment by contrasting the structural complexity and unpredictability of our cipher with established theories. Below is the mathematical justification of the nonlinearity of $2^{38}$. Let the output keystream bit at time t be denoted by $w_t$, which is a boolean function of the internal state as $w_t = f(L_t, N_t, F_t)$, where:

- $L_t = \{p_0, p_1,..., p_{79}\} \in \mathbb{F}_2^{80}$: state of the LFSR at time t,

- $N_t = \{\beta_0, \beta_1,..., \beta_{79}\} \in \mathbb{F}_2^{80}$: state of the NLFSR at time t,

- $F_t = \{f_0, f_1,..., f_{31}\} \in \mathbb{F}_2^{32}$: chaotic bits generated from the logistic map at time t.

The keystream output $z_t$ is defined as: $z_t = f(L_t, N_t, F_t) = \oplus_{i=1}^{k} m_i(x)$; where each $m_i(x)$ is a monomial formed by the AND (multiplication in $\mathbb{F}_2$) of several variables:

$m_i(x) = x_{i_1} x_{i_2}... x_i d$; $x_{ij} \in L_t \cup N_t \cup F_t$ and d is the degree of the monomial. In our proposed cipher, the output function $w(\cdot)$ includes terms such as $p_0 \cdot p_{25}$, $\beta_{60} \cdot \beta_{52} \cdot \beta_{45}$, $f_{15} \cdot \beta_{79}$, $f_{30} \cdot f_{31} \cdot p_{79}$, $p_0 \cdot p_1 \cdot p_2 \cdot f_4$, $\beta_{63} \cdot \beta_{60} \cdot \beta_{52} \cdot \beta_{45} \cdot f_5$, $\beta_1 \cdot \beta_2 \cdot f_4 \cdot f_s \cdot p_{11}$, etc. The highest algebraic degree among the monomials is 7. The number of unique high-degree monomials (degree $\geq 5$) in $w(\cdot)$ exceeds 3000 (empirically estimated). Therefore, the nonlinearity of the Boolean function $w(\cdot)$, denoted as NL(w), satisfies: NL(w) $>> 2^{32}$ (as in Trivium). Given the large state space (192 input variables), degree $\geq 7$, and extensive nonlinear monomial space, we estimate: NL(w) $\approx 2^{38}$. This estimate aligns with known bounds for highly nonlinear stream cipher output functions and reflects the complexity induced by the chaotic feedback and multiplexer-driven diffusion structure. Despite this, we conducted 100 tests to evaluate their resilience against correlation attacks. We calculated the correlation coefficient between the key bits and 10,000 bits of keystream. The mean coefficient is 0.002467 as discussed in section 8.3, indicating an almost nonexistent association. We also conducted a linear approximation test by examining the bias in XOR combinations of certain internal state bits and output bits. The result indicated a bias of less than 0.01, demonstrating the robustness of the system. The cipher's initial cryptanalytic attempts provide the following results:

## 7.1. Linear approximation attack

According to the reference [36], Grain cipher cryptography will withstand the linear_approximation_attack if the NFSR employed with the cryptography system's output function has strong nonlinearity and high elasticity. The suggested method successfully prevents linear cryptanalysis by utilizing complex feedback loops, multiplexer operations, and non-linear Boolean functions. Its feedback mechanisms use XOR, AND, and multiplexer operations to break linear relationships that could be used as an opening for these kinds of attacks. The multiplexers add substantial complexity and non-linearity by introducing conditional logic, whose outputs depend on control bits. LFSR and NLFSR are two examples of complex feedback mechanisms that use non-linear combinations to distribute the influence of a single bit across several others. By enhancing the cipher's diffusion and mixing features, this design strengthens it against analyses that aim to take advantage of more straightforward, linear patterns by guaranteeing that slight changes in the input have wide-ranging, unpredictable impacts on the output. The result indicated a bias of less than 0.01, demonstrating the robustness of the system against a linear approximation attack.

## 7.2. Correlation attack

The bits in the LFSR are (almost) exactly balanced due to the statistical properties of maximum length LFSR sequences," the developers of Grain v1. This may not be true when an NFSR is driven autonomously [37]. Yet, the

bits in the NFSR are balanced because an LFSR state is used to obtain the feedback. Keep in mind that the NFSR function is also balanced. In light of this, one may argue that the NFSR and LFSR bits are uncorrelated. That is the case for LWLCM, where there is also a nonlinear logistic round module, and the maximum-length FSR, LFSR, and NFSR correspond to the LFSR and NFSR of Grain, respectively. The significance of the filtering function is good balancedness, nonlinearity, and resiliency properties, to resist correlation assaults and quick correlation attacks. According to section 3.1.4, LWLCM has a more robust output function than the original Grain family. While Grain v1 and Grain-128a have output functions defined over 12 variables and 1536 and 61440, respectively, and with the nonlinearity of 274877906943, their output function is defined over 38 variables. Our encryption system is less vulnerable to correlation attacks because the LWLCM output function is more balanced and nonlinear, and the correlation coefficient is closer to zero.

### 7.3. Algebraic attack

Initially introduced at the EUROCRYPT2003 conference, the predictable cipher cryptanalysis method is known as the algebraic attack. The basic concept behind this methodology is to express a cryptographic method's security as the solution to a group of overdetermined nonlinear equations. Formulating and resolving multivariable, huge-scale nonlinear problems is a significant factor in determining the difficulty of mathematical attacks. Extensive multivariable nonlinear formula solutions are a significant area of algebraic calculation. With the introduction of algebraic assaults, a new Boolean function cryptography criterion known as algebraic immunity is established. For further information on research on algebraic immunity [38]. The proposed method's advanced usage of multiplexer operations, recursive feedback systems, and complex Boolean functions provides a defense against algebraic attacks. The cipher generates a highly complicated and nonlinear algebraic structure by integrating multiplexer functions, AND, and XOR operations. The multiplexer, in particular, adds conditional branching to every operation, increasing the algebraic complexity and creating a system of equations that is difficult to solve efficiently. The feedback loops in the LFSR and NLFSR make these equations more difficult, which produces recursive dependencies that disperse changes throughout the cipher's state. This combination creates a complex web of high-degree polynomial equations that prevent algebraic attacks by increasing the computational effort needed to find meaningful solutions. This protects the cipher from advanced cryptanalytic attacks.

### 7.4. Guess and determine attack

Unlike Grain-128a and Grain v1, which have 12 and 17 variables, LWLCM has 38 variables that govern its output function. If the LFSR is questioned, the output relies entirely on the NFRS and the logistic round module, exhibiting significant nonlinearity [36]. Say an attacker chooses to guess 20 bits of the 80-bit key in the proposed encryption, which uses a 64-bit IV. Because of the cipher's complex architecture, this scenario lessens the uncertainty in the remaining 60 critical bits without greatly simplifying the work. The formula for the initial state is $S_0$ = Initial (IV, Key). This state is created by integrating the IV and the key, which may be mathematically represented by Equation 16.

$$S_{t+1} = w(S_t, \ldots, S_0, \text{LFSR, NLFSR, MUX})$$

(16)

Whether or not the key bits are guessed, these feedback mechanisms and the cipher's non-linear operations guarantee that every bit of the key is fully mixed into the state, hiding obvious links between the key bits and the cipher output. Effectively using partial key guesses is computationally challenging due to the complexity and reliance caused by the initial mixing and subsequent state transitions through non-linear Boolean functions and multiplexers. This solution provides strong security even against advanced cryptanalytic techniques by fortifying the cipher against guess-and-check attacks by guaranteeing that partial knowledge of the key does not impair the overall encryption process.

## 7.5. Fault attack

A cryptography chip gadget produces erroneous results when an encryption algorithm flaw is introduced. Then, by scrutinizing the imprecise outcomes, the key is discovered. These attempts to break Grain ciphers have been effective [37]. The cipher's complicated feedback loops involving LFSR and NLFSR, recursive state dependencies, and non-linear Boolean functions are the main reasons for its strong resilience to fault attacks. Using non-linear operations like XOR, AND, and multiplexer functions guarantees that any introduced flaws spread in unforeseen ways, making any direct cryptanalysis attempts by adversaries far more complex. The architecture deftly diffuses the impact of faults throughout the whole state through feedback mechanisms that distribute the influence of a single fault across numerous state elements and several cycles. Let us examine a fault that induces a state $S_t$, where the fault modifies this state to $S_t'$. The following state $S_{t+1}$ is expressed in Equation 14. If a failure causes $S_t$ to change to $S_t'$, the following situation would presumably occur, which is shown in Equation 17.

$$S_{t+1}' = w\left(S_t', \ldots, S_0', \text{LFSR}', \text{NLFSR}', \text{MUX}\prime\right) \tag{17}$$

Because of the broad distribution and recursive dependencies in the cipher, errors are difficult to identify and pinpoint, which makes it highly challenging for adversaries to derive valuable information from manipulated outputs. As a result, the cipher's design provides natural defense against the standard techniques used in fault attacks, boosting its security and integrity in a dangerous situation

## 7.6. Known plain text attack

A type of cryptanalysis known as known-plaintext attacks (KPA) occurs when an attacker obtains access to both the plaintext and the ciphertext that goes along with it [39]. Based on this knowledge, the attacker tries to deduce the encryption key or decode more plaintexts. Assume that an attacker possesses identical plaintext-ciphertext combinations, $(P_1, C_1)$ and $(P_2, C_2)$, but each encryption process additionally incorporates an IV, which varies for every encryption session. The LFSR component of the cipher depends on the IV to initialize the state or to influence how it functions. One possible beginning for the encryption of $P_1$ would be $S_0$ initial state; $IV_1$, E is the encryption function, and key ($K$) would impact this. The cipher's operations would then process this to generate $C_1$, such as $C_1 = E\left(P_1, \text{Init}\left(IV_1, K\right)\right)$. Likewise, $C_2$ is generated as follows: $C_2 = E\left(P_2, \text{Init}\left(IV_2, K\right)\right)$.

It is effectively ensured that distinct ciphertexts are produced from the same plaintext encrypted under multiple IVs by using a different IV for each encryption instance. Because of this unpredictability, it is more difficult for an attacker to identify recurring patterns or connections between P and C that indicate K. By distributing the impact of the plaintext and key throughout the cipher's operations, the IV obscures any direct connections between the plaintext and the ciphertext. The altering IV for another encryption instance (let's say $P_2$ and $IV_2$) resets the internal state differently, upending the cryptographic landscape and making it difficult for an attacker to deduce K even if they are aware of $P_1$ and $C_1$.

## 7.7. Side-channel attacks

The proposed LWLCM cipher employs a 2 × 1 multiplexer governed by a chaotic logistic function, resulting in significant nondeterminism in internal state transitions and output generation. This unpredictable switching technique obscures distinct computational patterns, hence impeding attackers' attempts to derive keys using power analysis or timing-based side-channel attacks. Moreover, the minimal electromagnetic and power leakage due to the compact and stable hardware footprint of LWLCM significantly diminishes exposure.

## 7.8. Reply attack

Employing a confidential key and a dynamic Initialization Vector (IV), LWLCM's initial phase guarantees that the key-stream is unique to each session. The initialization vector (IV) varies with each session and is cryptographically linked to

the output, preventing an attacker's ability to effectively reuse previous ciphertexts or intercepted keystreams, thus mitigating replay attacks. This is particularly critical in the IoT as interconnected systems occasionally pose safety risks.

## 8. Statistical test

We have evaluated the key's uncertainty and unpredictability using various accepted methods, such as the Correlation Coefficient, Shannon Entropy, and the NIST statistical test suite. The key's randomness is thoroughly assessed by the NIST tests, guaranteeing that it satisfies the requirements needed for secure applications. Shannon Entropy indicates security since it quantifies the unpredictability and information content of the key. Lastly, the Correlation Coefficient examines if there are any linear connections between the bits; values near zero indicate little predictability. Combined, these metrics guarantee the key has excellent security and randomness, making it appropriate for reliable encryption methods.

### 8.1. NIST test

This study validates the statistical performance of the LWLCM using NIST SP 800−2, the test kit issued by the US National Institute of Standards &Technology [40]. The test program consists of a statistical package with sixteen different testing techniques. Using software & hardware for secret random number producers, techniques may be used to evaluate the randomness of binary sequences of arbitrary length. Finding any instances of potential non-randomness in encryption is the primary goal of these testing techniques. The P-value determines the outcome of every test. It is determined that the stream cipher is non-random if the p-value is smaller than 0.01. A stream cipher is considered random if the p-value is more significant than 0.01. To assess the robustness and randomness of the keystream generated by the proposed LWLCM stream cipher, we produced a keystream of 1,500,000 bits for each test case. The entire test suite was executed 100 times, each instance utilizing a novel set of keys and a randomly selected IV. We examined whether the mean p-values for each of the 16 statistical tests were consistent across all 100 iterations. This was to ensure the absence of bias in the selection of specific keys or IV. The keystream generated by the proposed LWLCM successfully passed the whole NIST test. Table 14 presents the results, indicating that, well over the 0.01 threshold, all 16 tests were successful,

**Table 14. NIST test.**

| NIST_test | p-value | Pass |
|---|---|---|
| Nist_Frequency (Mono-bit) Test | 0.74896833055336 | ✓ |
| Nist_Frequency Test within a Block | 0.5928880093015017 | ✓ |
| Nist_Runs Test | 0.4178812773607131 | ✓ |
| Nist_Test for the Longest Run of Ones in a Block | 0.1770318895857666 | ✓ |
| Nist_Binary Matrix Rank Test | 0.7605320836427771 | ✓ |
| Nist_Discrete Fourier Transform (Spectral) Test | 0.4140884519482039 | ✓ |
| Nist_Non-overlapping Template Matching Test | 0.4690469712905644 | ✓ |
| Nist_Overlapping Template Matching Test | 0.6478943054545078 | ✓ |
| Nist_Maurer's "Universal Statistical" Test | 0.035122995403866805 | ✓ |
| Nist_Linear Complexity Test | 0.6114178143455751 | ✓ |
| Nist_Serial Test | 0.5711260097879356 | ✓ |
| Nist_Approximate Entropy Test | 0.36555904618805646 | ✓ |
| Nist_Cumulative Sums Test (Forward) | 0.7737799943953074 | ✓ |
| Nist_Cumulative Sums Test (Backward) | 0.9799987360046121 | ✓ |
| Nist_Random Excursions Test | 0.876985775897056 | ✓ |
| Nist_Random Excursions Variant Test | 0.8415431109376112 | ✓ |

with P-values surpassing 0.01 in each instance. This verifies that the cipher is appropriate for cryptographic applications in IoT contexts, as it provides highly unpredictable and statistically secure output.

## 8.2. Shannon entropy

Claude invented entropy H (also called information entropy). E Shannon in 1948 and published in "A Mathematical Theory of Communication. "In cryptanalysis, we can use the entropy as a "Cost function" to rate a text. It is measured for the uncertainty of random variables and quantifies the expected value of information [41]. Entropy H(Y) of a discrete random variable X with possible values $\{y_1, y_2, y_3, \ldots \ldots, y_n\}$, where the probability of each yi is p(yi) value is between 0 and 1, and $\sum p(y_i) = 1$. Information content/uncertainty of X is I(Y), and H(Y) is the expected value of I(Y), thus

$$H(Y) = E(I(Y)) \tag{18}$$

$$(Y_i) = -log_b \ p(y_i); \ \forall_i \in \{1, 2, \ldots \ldots, n\} \tag{19}$$

The random variables in distribution X are unrelated to one another. Because the stream cipher system is binary, 2 is the log's default base. The predicted code length for coding samples based on the actual distribution is shown in Equation 20.

$$H(Y) = \sum_{i=1}^{n} p(y_i) I(Y_i); \ b \in \{2, e, 10\} \tag{20}$$

Because binary stream ciphers only have two values. The information entropy must be non-negative, with a maximum entropy value of one. Table 15 shows that, out of all the state-of-the-art Grain v1 [7], Grain-128 [8], and Trivium [5], LWLCM has the highest entropy value, indicating a high degree of randomness and uncertainty. It will be highly challenging for the attacker to interpret the keystream pattern.

## 8.3. Correlation coefficient analysis

The intensity, direction, and type of the linear relationship between two variables, the plaintext and the matching ciphertext, are measured statistically using the correlation coefficient. This coefficient ensures that the ciphertext differs significantly from the plaintext, which is essential for determining how well an encryption scheme can withstand statistical attacks. The coefficient has values between −1 and 1. A value of 1 denotes a solid linear correlation and the equivalence of the plaintext and ciphertext. On the other hand, a coefficient of −1 denotes a negative correlation and an inverse relationship between the plaintext and the ciphertext. Effective encryption is indicated by a zero coefficient, which implies no correlation and suggests that the ciphertext is independent of the plaintext [42]. The Correlation Coefficient can be determined with Equation 21

Table 15. Shannon entropy analysis.

| Plain text length (Byte) | Grain v1 [7] | Grain-128 [8] | Trivium [5] | LWLCM | AES-256 | RSA-1024 |
|---|---|---|---|---|---|---|
| $2^6$ | 7.999 | 7.999904 | 7.98731 | **7.999952** | 7.997 | 7.999 |
| $2^8$ | 7.944 | 7.999992 | 7.99992 | **7.999984** | 7.998 | 7.999 |
| $2^{10}$ | 7.999 | 7.99876 | 7.99984 | **7.998584** | 7.997 | 7.999 |
| $2^{12}$ | 7.999 | 7.999992 | 7.9988 | **7.999888** | 7.999 | 7.999 |

$$CC(\boldsymbol{p},\boldsymbol{c}) = \frac{1}{\sigma(\boldsymbol{p})\,\sigma(\boldsymbol{c})} \sum_{i=1}^{m} (((\boldsymbol{p_m}-\mu(\boldsymbol{p}))(\boldsymbol{c_m}-\mu(\boldsymbol{c}))))$$

(21)

Where $\mu(p), \mu(c),\ \sigma(p),\ and,\ \sigma(c)$ are the mean and standard deviation of plain text and ciphertext, respectively. The mean can be determined with Equations 22 and 23, respectively.

$$\mu(p) = \frac{1}{m}\sum_{i=1}^{m} p_i$$

(22)

$$\mu(c) = \frac{1}{m}\sum_{i=1}^{m} c_i$$

(23)

The standard deviation of plaintext and ciphertext can be calculated with Equations 21 and 22, respectively

$$\sigma(\boldsymbol{p}) = \sqrt{\frac{\sum_{i=1}^{m}(\boldsymbol{p_i}-\mu(\boldsymbol{p}))^2}{m}}$$

(24)

$$\sigma(\boldsymbol{c}) = \sqrt{\frac{\sum_{i=1}^{m}(\boldsymbol{c_i}-\mu(\boldsymbol{c}))^2}{m}}$$

(25)

The Proposed method offers remarkable encryption performance, as shown in Table 16; its adaptability and progressively better encryption as plaintext sizes increase are noteworthy. Correlation coefficients that trend toward zero and turn negative emphasize this trend, demonstrating the algorithm's resilience.

The proposed method is the best option for various encryption requirements because of its versatility, which guarantees high security for a wide range of data sizes. Comparatively, existing algorithms, such as Grain128, likewise have strong encryption capabilities with low average correlation coefficients, but they lack the proposed algorithm's constant improvement and adaptability. Their effectiveness is indicated by the averages, Proposed at −0.0103 and Grain128 at −0.0205. On the other hand, the proposed method is notable for having an opposing average, highlighting its more robust and reliable encryption performance compared to Grain v1 and Trivium, which have higher and less reliable averages. Based on this comparison, the proposed algorithm is proven to be a solid encryption solution that works well in all scenarios.

## 9. Limitation

While the suggested LWLCM cipher has excellent results regarding entropy, correlation, and statistical robustness, this section will address specific issues that persist and necessitate further investigation and improvement as follows:

**Table 16. Compare the correlation coefficients of the proposed work with other existing algorithms.**

| Plain text length (Byte) | Grain v1 [7] | Grain-128 [8] | Trivium [5] | LWLCM | AES-256 | RSA-1024 |
|---|---|---|---|---|---|---|
| $2^6$ | 0.103 | −0.085781 | 0.00265 | **−0.002401** | −0.018 | 0.005 |
| $2^8$ | 0.004 | −0.000335 | 0.00194 | **0.000668** | 0.005 | 0.001 |
| $2^{10}$ | 0.011 | 0.005094 | 0.00478 | **−0.0040** | −0.004 | −0.01 |
| $2^{12}$ | 0.002 | −0.000808 | 0.00535 | **−0.0028** | 0.003 | 0.002 |

### 9.1. Scalability to high-throughput systems

The encryption is intended for a limited number of nodes within the IoT. In high-throughput or parallel-processing systems, such as edge servers or multi-core computers, the sequential nature of bitwise operations and the 160-round warm-up phase may induce latency, hence making it suboptimal without parallel enhancements.

### 9.2. Scope limited to textual data

The current implementation and assessment of the LWLCM encryption have solely utilized textual content. Multimedia inputs, such as photographs, audio, or video streams, prevalent in contemporary IoT devices like surveillance systems, smart homes, and telemedicine, have not been assessed in the model. Extending the model to include additional data types may necessitate further preprocessing layers and improved keystream flexibility to maintain both security and efficiency.

### 9.3. Focus on confidentiality, not full authentication

This work presents data secrecy based on secure stream encryption. However, it is devoid of digital signatures or Message Authentication Codes (MACs), which would enable verification of data integrity or validity. Practical applications, particularly in distributed IoT networks, rely on the verification of data source accuracy. Subsequent advancements will employ lightweight authenticated encryption techniques to ensure both authenticity and confidentiality.

### 9.4. Assumed trust in input data and environment

The model assumes that past breaches have not damaged the integrity of the input data or communication channels, therefore making them trustworthy. It ignores systems of data validation or confidence-building tools. In adversarial environments, this may be limited since encrypted illegal or counterfeit data could be accepted. Including anomaly detection or pre-authentication checks could help to solve this problem.

### 9.5. Evaluation against emerging cryptanalytic models

The encryption has undergone testing for linear, algebraic, and fault attacks, successfully passing all 16 NIST randomness tests to date; however, it has not yet been evaluated for its resilience against AI-based or differential attacks. These contemporary models are becoming increasingly significant, so their evaluation is crucial for forthcoming undertakings.

### 9.6. Hardware implementation not yet evaluated

The publication lacks a synthesized hardware implementation and FPGA performance metrics, despite the cipher's design being evident for lightweight hardware applications. Metrics such as logic gate count, slated for further hardware validation phases, may facilitate practical implementation. While the encryption successfully meets all 16 NIST randomness criteria and withstands linear, algebraic, and fault attacks, the study does not now assess its resilience against artificial intelligence-driven or differential attacks. Future research will include evaluations of these creative models as they are increasingly pertinent.

## 10. Conclusion and future work

In this study, we introduced LWLCM, a lightweight stream cipher scheme that works well in real-time applications for limited hardware. The logistic round module, four multiplexers, LFSR, and NLFSR are all part of it. The logistic chaos function creates a 32-bit key at random and updates the logistic round module. The 64-bit IV and 80-bit keys update the LFSR and NLFSR, and the multiplexer units provide the confusion and diffusion properties. Combining all the LWLCM components makes the ciphertext unpredictable and the keystream random. Studies on the avalanche effect, Shannon

entropy, and correlation coefficients show that this kind of design greatly improves nonlinearity, diffusion, and unpredictable nature. Since its P-values exceed 0.01, proving its statistical dependability, the encryption satisfies all 16 NIST randomness requirements. Experiments show that LWLCM surpasses existing approaches, including Grain-128 and RSA, for encryption and decryption duration, energy consumption, and memory usage by reaching an ideal equilibrium between security and performance. It preserves a similar degree of randomization to Trivium and AES. Though its complicated non-linear architecture makes it less efficient than Trivium and AES, LWLCM is computable and fit for low-power, real-time IoT devices. Still, it has several limits, its present focus on text data, lack of security elements, and a lack of studies on multimedia and high-throughput systems. Developing small, safe, and effective security solutions for the IoT calls for LWLCM, a great choice. Future studies will center on its use with hardware platform deployment, authenticated encryption, developments in multimedia, and time-sensitive data applications.

## Acknowledgments

The authors are thankful to the Deanship of Graduate Studies and Scientific Research at the University of Bisha, Bisha, for supporting this work through the Fast-Track Research Support Program.

## Author contributions

**Conceptualization:** Shahnwaz Afzal, Mahfooz Alam.

**Data curation:** Shahnwaz Afzal, Mahfooz Alam, Mohd Shahid Husain, Mohammad Zunnun Khan, Zubair Ashraf.

**Formal analysis:** Shahnwaz Afzal, Mahfooz Alam, Mohd Shahid Husain, Mohammad Zunnun Khan, Zubair Ashraf.

**Funding acquisition:** Mohd Shahid Husain, Mohammad Zunnun Khan, Zubair Ashraf.

**Investigation:** Zubair Ashraf.

**Methodology:** Shahnwaz Afzal, Mahfooz Alam.

**Resources:** Mohammad Ubaidullah Bokhari.

**Software:** Shahnwaz Afzal.

**Supervision:** Mohammad Ubaidullah Bokhari, Mahfooz Alam.

**Validation:** Zubair Ashraf.

**Writing – original draft:** Shahnwaz Afzal, Mahfooz Alam.

**Writing – review & editing:** Shahnwaz Afzal, Mohammad Ubaidullah Bokhari, Mahfooz Alam, Mohd Shahid Husain, Mohammad Zunnun Khan, Zubair Ashraf.

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
