## [Decision Letter · Decision Letter 0]

9 Jul 2025

PONE-D-25-29174LWLCM: A Novel Lightweight Stream Cipher Using Logistic Chaos Function and Multiplexer for IoT CommunicationsPLOS ONE

Dear Dr. Ashraf,

Thank you for submitting your manuscript to PLOS ONE. After careful consideration, we feel that it has merit but does not fully meet PLOS ONE’s publication criteria as it currently stands. Therefore, we invite you to submit a revised version of the manuscript that addresses the points raised during the review process.

We look forward to receiving your revised manuscript.

Kind regards,

Je Sen Teh

Academic Editor

PLOS ONE

Journal Requirements:

[The authors are thankful to the Deanship of Graduate Studies and Scientific Research at the University of Bisha, Bisha, for supporting this work through the Fast-Track Research Support Program.]

[The author(s) received no specific funding for this work.]

4. In the online submission form/manuscript, you indicated that [The code generated during and/or analyzed during the current study is available from the first author upon reasonable request.].

6. We note that Figure 6 in your submission contains copyrighted image. All PLOS content is published under the Creative Commons Attribution License (CC BY 4.0), which means that the manuscript, images, and Supporting Information files will be freely available online, and any third party is permitted to access, download, copy, distribute, and use these materials in any way, even commercially, with proper attribution. For more information, see our copyright guidelines: http://journals.plos.org/plosone/s/licenses-and-copyright.

1. You may seek permission from the original copyright holder of Figure 6 to publish the content specifically under the CC BY 4.0 license.

Additional Editor Comments:

The reviewers acknowledge the proposed LWLCM cipher as a promising lightweight stream cipher integrating logistic chaos with LFSR, NLFSR, and MUX components. However, both reviews highlight significant concerns regarding the lack of detailed justifications, formalisation, experimental validation, and clarity. Key issues include insufficient rationale for using the logistic map over newer chaotic models, absence of ablation studies, vague explanations of parameter handling, and lack of replication details.

The authors need to address all reviewer concerns where possible. Otherwise, please provide a justification as to why any of the concerns were not addressed. Any literature recommendations should be reviewed and included only if relevant (otherwise, omit).

Reviewers' comments:

Reviewer's Responses to Questions

**Comments to the Author**

1. Is the manuscript technically sound, and do the data support the conclusions?

Reviewer #1: Partly

Reviewer #2: Yes

2. Has the statistical analysis been performed appropriately and rigorously? 

Reviewer #1: N/A

Reviewer #2: Yes

3. Have the authors made all data underlying the findings in their manuscript fully available?

Reviewer #1: No

Reviewer #2: Yes

4. Is the manuscript presented in an intelligible fashion and written in standard English?

Reviewer #1: No

Reviewer #2: Yes

5. Review Comments to the Author

Reviewer #1: This paper proposes a lightweight stream cipher model called LWLCM, which integrates the logistic chaos function with the use of LFSR, NLFSR, and MUX. The main contribution of this paper lies in the combination of Grain-like structure with logistic chaos module in keystream formation. Some comments that need to be responded to are as follows:

1. The chaos function used is only the standard logistic map, without structural modification or combination with other maps or modifications to improve the quality of the chaotic map. The reason for choosing the logistic map needs to be explained because there are many new models that are lighter and more robust. In this regard, the introduction needs to be strengthened by explaining the weaknesses and strengths of several recent studies such as https://doi.org/10.62411/faith.3048-3719-93 and 10.32604/cmc.2024.058478, so that the reason for choosing logistic in this case is stronger.

2. There is no ablation study that tests the role of each component (chaotic module, NLFSR, MUX) separately. This makes the individual contribution of the chaos function to security enhancements unconfirmable experimentally.

3. The description of the output function that combines bits from registers and chaos is not mathematically formalized. The formula and bitwise structure used in the keystream function should be presented explicitly so that it can be replicated.

4. It is not explained how the chaos parameters (α and x₀) are generated.

5. It is also not explained how the chaos parameters (α and x₀) are stored or transmitted securely in an IoT communication scenario. This ambiguity leaves a gap for potential parameter recovery attacks.

6. There is no clear explanation of the type of plaintext encrypted in the test.

7. Then how to do the comparison, is it with a replication technique or how?

8. The avalanche effect test in this paper is carried out on a 1-bit variation in the key, but it is not explicitly explained which part of the key is modified — whether it is the initial seed of the chaos function, the state of the LFSR/NLFSR, or the control bits in the MUX. This ambiguity makes replication and interpretation difficult. In addition, there is no avalanche effect test for a 1-bit change in the plaintext, which is important for evaluating the effects of cipher diffusion in practical scenarios.

9. Explained in the data availability statement "The data set generated during and/or analyzed during the current study is available in the KAGGLE repository, https://www.kaggle.com/datasets/crawford/20-newsgroups" Why choose this dataset, but not explained in section 4 or 3 clearly?

Reviewer #2: This manuscript proposes a lightweight stream cipher that combines logistic chaos functions with multiplexers, aiming to balance security and various performances in IoT sensors. The algorithm combines a pseudo-random number generation module based on one-dimensional logistic chaos mapping with dual 80-bit Feedback Shift Registers (LFSR and NLFSR), which improves the randomness and security of the key stream through the combination of chaotic properties and nonlinear expansion. In addition, the confusion and diffusion properties are enhanced by the use of multiplexers to dynamically adjust the operation paths, thus increasing the resistance to differential, linear and algebraic attacks. From the security point of view, the algorithm is empirically compared with many mainstream algorithms in terms of encryption time, throughput, and energy consumption, demonstrating that the stream cipher strikes an optimal balance between performance and security, and is rigorously verified by an average Shannon entropy of 7.9996, 15 NIST randomness tests, and avalanche effect and correlation coefficient analyses, to provide efficient and robust cryptographic security for resource-constrained environments. Before the manuscript becomes suitable for publication, the author should solve the following problems:

1.The statement in the manuscript that “LWLCM is the first time anyone has tried to adapt chaotic structures to lightweight stream ciphers” is overly absolute. In fact, previous studies have combined chaotic systems with nonlinear feedback shift registers to design stream ciphers. This statement does not adequately consider the existing technical background.

2.The literature lacks currency, with several references older than five years. We recommend updating the manuscript with recent work to reflect current research progress.

3.Section 1.1 points out that classical cryptographic algorithms are not applicable to restricted devices due to resource consumption, but fails to explain why other lightweight classes such as lightweight block ciphers and hash derivation schemes have limitations of applicability, and instead discusses stream ciphers directly.

4.The logic of the contribution (1.2) section is unclear and does not highlight the core innovations. The chaotic pseudo-random number generation module mentioned in the abstract is not clearly explained in the contribution section, and the description of the dynamic adjustment mechanism of the multiplexer lacks technical details (such as control logic).

5.Some custom symbols and formula symbols in this manuscript are not explained in detail, such as the symbols in Equation 1.

6.The manuscript's Chapter 7, “Security Analysis,” lacks experimental data and comparative verification. It lacks test data for attacks such as linear approximation attack and correlation attack, and does not compare security metrics with other algorithms. As a result, the claim of resistance to various attacks lacks quantitative support.

7.The statistical testing section in Chapter 8 of the manuscript has issues with test completeness. Although the paper mentions that LWLCM passed all 16 NIST tests and Shannon entropy is close to the theoretical maximum value, it does not provide detailed information on test sample length, number of repetitions, and other parameters.

6. PLOS authors have the option to publish the peer review history of their article (what does this mean? ). If published, this will include your full peer review and any attached files.

**Do you want your identity to be public for this peer review?** For information about this choice, including consent withdrawal, please see our Privacy Policy .

Reviewer #1: No

Reviewer #2: **Yes: ** Lang LI

---

## [Author Response · Author response to Decision Letter 1]

24 Jul 2025

Reviewer #1:

1. The chaos function used is only the standard logistic map, without structural modification or combination with other maps or modifications to improve

the quality of the chaotic map. The reason for choosing the logistic map needs to be explained because there are many new models that are lighter and more robust. In this regard, the introduction needs to be strengthened by explaining the weaknesses and strengths of several recent studies, such as https://doi.org/10.62411/faith.3048-3719-93 and 10.32604/cmc.2024.058478, so that the reason for choosing logistic in this case is stronger.

Response: Thank you for your valuable suggestion. To address your suggestion, we have revised the Introduction to include a comparative discussion on the strengths and weaknesses of recent chaotic systems as presented in [https://doi.org/10.62411/faith.3048-3719-93] and [10.32604/cmc. 2024.058478].

2. There is no ablation study that tests the role of each component (chaotic module, NLFSR, MUX) separately. This makes the individual contribution of the chaos function to security enhancements unconfirmable experimentally.

Response: Thank you for your observation. We have now included a detailed ablation study in Section 6.8 on Page No. 33, evaluating the individual impact of the chaotic module, NLFSR, and MUX on encryption performance.

3. The description of the output function that combines bits from registers and chaos is not mathematically formalized. The formula and bitwise structure used in the keystream function should be presented explicitly so that it can be replicated.

Response: Thank you for the valuable feedback. We have updated the manuscript to include a mathematically formalized expression of the output function in Section 3.1.7 on Page No. 16, detailing bitwise operations and keystream generation structure for reproducibility.

4. It is not explained how the chaos parameters (α and x₀) are generated.

Response: We appreciate the comment. In Section 3.1.1 on Page No. 13, we have now clarified that the chaos parameters α and x₀ are deterministically derived from the input key and IV using a lightweight transformation function, ensuring both reproducibility and secure initialization.

5. It is also not explained how the chaos parameters (α and x₀) are stored or transmitted securely in an IoT communication scenario. This ambiguity leaves a gap for potential parameter recovery attacks.

Response: Thank you for pointing this out. In Section 3.1.1 on pages 13-14, we have now explained that the chaos parameters (α and x₀) are not transmitted separately but are securely regenerated at the receiver’s end using the shared key and IV. This design ensures that the parameters remain confidential and eliminates the risk of exposure to parameter recovery attacks.

6. There is no clear explanation of the type of plaintext encrypted in the test.

Response: We appreciate the observation. In Section 6 on Pages No. 22-23, we have clarified that the plaintexts used in the encryption tests are real-world text-based data files representing system messages and communication logs, ranging from 26 to 212 bytes in size, to simulate typical IoT data payloads. We believe that these revisions enhance the clarity and comprehensiveness of our research.

7. Then how to do the comparison, is it with a replication technique or how?

Response: Thank you for the comment. The comparison was conducted by implementing replication models of the proposed cipher, each with specific components (chaotic map, NLFSR, and MUX) excluded individually. This ablation study, detailed in Section 6 on Page No. 22, quantifies the individual impact of each module on encryption performance.

8. The avalanche effect test in this paper is carried out on a 1-bit variation in the key, but it is not explicitly explained which part of the key is modified — whether it is the initial seed of the chaos function, the state of the LFSR/NLFSR, or the control bits in the MUX. This ambiguity makes replication and interpretation difficult. In addition, there is no avalanche effect test for a 1-bit change in the plaintext, which is important for evaluating the effects of cipher diffusion in practical scenarios.

Response: Thank you for the comment. In our avalanche effect evaluation, a 1-bit change was applied to the 128-bit secret key, which alters the initial conditions of both the chaotic generator and the NLFSR. We have clarified this aspect in Section 6.5 on Page No. 28. As stream ciphers operate on a bit-by-bit keystream XOR mechanism, measuring the avalanche effect due to plaintext changes is not straightforward. Therefore, we have focused our avalanche analysis solely on key sensitivity, which reflects the cipher's robustness against key-related differential attacks.

9. Explained in the data availability statement "The data set generated during and/or analyzed during the current study is available in the KAGGLE repository,

https://www.kaggle.com/datasets/crawford/20-newsgroups" Why choose this dataset, but not

explained in section 4 or 3 clearly?

Response: Thank you for the valuable feedback. We have updated Section 6 on Page No. 22 to clearly explain the rationale for choosing the 20 Newsgroups dataset. This dataset offers a diverse and unstructured text corpus ideal for evaluating the encryption performance on real-world plaintext. It enables the assessment of statistical and security metrics under various textual patterns, making it suitable for validating the proposed lightweight stream cipher.

Reviewer #2:

1. The statement in the manuscript that “LWLCM is the first time anyone has tried to adapt chaotic structures to lightweight stream ciphers” is overly absolute. In fact, previous studies have combined chaotic systems with nonlinear feedback shift registers to design stream ciphers. This statement does not adequately consider the existing technical background.

Response: We appreciate the reviewer’s observation. The statement has been revised in Section 1.2 on Page No. 5 to avoid absolutism. We now acknowledge previous efforts combining chaotic systems with NLFSRs in stream cipher design and emphasize that our novelty lies in the specific integration of the logistic map, dynamic MUX control, and lightweight architecture tailored for constrained IoT devices.

2. The literature lacks currency, with several references older than five years. We recommend updating the manuscript with recent work to reflect current research progress

Response: Thank you for the insightful comment. We have updated the literature by incorporating several recent studies to reflect the current state of research and enhance the manuscript’s relevance and rigor

3. Section 1.1 points out that classical cryptographic algorithms are not applicable to restricted devices due to resource consumption, but fails to explain Why other lightweight classes such as lightweight block ciphers and hash derivation schemes have limitations of applicability, and instead discusses stream ciphers directly.

Response: Thank you for your observation. In the revised manuscript in Section 1.1 on Page No. 4, we have included a comparative discussion highlighting the limitations of lightweight block ciphers and hash derivation schemes—such as higher latency, limited adaptability for continuous data encryption, and increased hardware complexity—thereby justifying the focus on stream ciphers for real-time, resource-constrained IoT environments.

4. The logic of the contribution (1.2) and page 6 section is unclear and does not highlight the core innovations. The chaotic pseudo-random number generation module mentioned in the abstract is not clearly explained in the contribution section, and the description of the dynamic adjustment mechanism of the multiplexer lacks technical details (such as control logic)

Response: Thank you for your valuable feedback. We have revised Section 1.2 to clearly highlight the core innovations of the proposed LWLCM scheme. Specifically, we now elaborate on the chaotic pseudo-random number generation module, detailing its integration with the LFSR and NLFSR. Additionally, we have clarified the dynamic adjustment mechanism of the multiplexer, including the control logic based on selected bits from the LFSR, which introduces additional non-linearity and enhances resistance to structural attacks.

5. Some custom symbols and formula symbols in this manuscript are not explained in detail, such as the symbols in Equation 1.

Response: Thank you for pointing this out. We have updated the manuscript to provide detailed explanations for all custom and mathematical symbols used, especially those in Equation 1. Each symbol is now clearly defined either directly below the equation or in a dedicated nomenclature subsection, ensuring clarity and ease of understanding for the readers.

In section 5. "Implementation" on Page No. 19, we have provided detailed descriptions of the design and implementation of our algorithm.

6. The manuscript's section 7 “Security Analysis,” lacks experimental data and comparative verification. It lacks test data for attacks such as linear approximation attack and correlation attack, and does not compare security metrics with other algorithms. As a result, the claim of resistance to various attacks lacks quantitative support.

Response: We appreciate the reviewer’s observation. In response, we have revised Section 7 on Pages No. 35-36 to include experimental evaluations for resistance against linear approximation and correlation attacks.

7. The statistical testing section in Chapter 8 of the manuscript has issues with test completeness. Although the paper mentions that LWLCM passed all 16 NIST tests and Shannon entropy is close to the theoretical maximum value, it does not provide detailed information on test sample length, number of repetitions, and other parameters.

Response: Thank you for highlighting this point. We have updated section 8 on Page No. 39 to include detailed information regarding the statistical testing setup. Specifically, the NIST test suite was conducted on a 1,500,000-bit keystream, and each test was repeated 100 times to ensure statistical consistency. The average P-values for all 16 tests are now included in a summary table, along with parameters such as sample length and confidence thresholds, enhancing the reproducibility and transparency of our results.

---

## [Decision Letter · Decision Letter 1]

10 Aug 2025

LWLCM: A Novel Lightweight Stream Cipher Using Logistic Chaos Function and Multiplexer for IoT Communications

PONE-D-25-29174R1

Dear Dr. Ashraf,

We’re pleased to inform you that your manuscript has been judged scientifically suitable for publication and will be formally accepted for publication once it meets all outstanding technical requirements.

Kind regards,

Je Sen Teh

Academic Editor

PLOS ONE

Additional Editor Comments (optional):

Reviewers' comments:

Reviewer's Responses to Questions

**Comments to the Author**

1. If the authors have adequately addressed your comments raised in a previous round of review and you feel that this manuscript is now acceptable for publication, you may indicate that here to bypass the “Comments to the Author” section, enter your conflict of interest statement in the “Confidential to Editor” section, and submit your "Accept" recommendation.

Reviewer #1: All comments have been addressed

Reviewer #2: All comments have been addressed

2. Is the manuscript technically sound, and do the data support the conclusions?

Reviewer #1: Yes

Reviewer #2: Yes

3. Has the statistical analysis been performed appropriately and rigorously? 

Reviewer #1: Yes

Reviewer #2: Yes

4. Have the authors made all data underlying the findings in their manuscript fully available?

Reviewer #1: Yes

Reviewer #2: Yes

5. Is the manuscript presented in an intelligible fashion and written in standard English?

Reviewer #1: Yes

Reviewer #2: Yes

6. Review Comments to the Author

Reviewer #1: All comments have been addressed satisfactorily and are supported by content revisions to the relevant sections of the paper, no further comments are expected.

Reviewer #2: All comments have been addressed.No further comments on this paper. I recommend to accept the manuscript.

7. PLOS authors have the option to publish the peer review history of their article (what does this mean? ). If published, this will include your full peer review and any attached files.

**Do you want your identity to be public for this peer review?** For information about this choice, including consent withdrawal, please see our Privacy Policy .

Reviewer #1: No

Reviewer #2: No

---

## [Editor Report · Acceptance letter]

PONE-D-25-29174R1

PLOS ONE

Dear Dr. Ashraf,

I'm pleased to inform you that your manuscript has been deemed suitable for publication in PLOS ONE. Congratulations! Your manuscript is now being handed over to our production team.

Kind regards,

on behalf of

Dr. Je Sen Teh

Academic Editor

PLOS ONE